



**1    Modelling the diurnal and seasonal dynamics of soil CO$_2$ exchange in a semiarid ecosystem with**

**2    high plant-interspace heterogeneity**

Jinnan Gong[1], Ben Wang [1,2,], Xin Jia[1,2,], Wei Feng[2], Tianshan Zha[2], Seppo Kellomäki[1] and Heli
Peltola[1]
[1] School of Forest Sciences, University of Eastern Finland, P.O. Box 111, 80101 Joensuu, Finland
[2] Yanchi Research Station, School of Soil and Water Conservation, Beijing Forestry University,
Beijing 100083, China
Correspondence to: Jinnan Gong (jinnan.gong@uef.fi)
Abstract
This study represents a first attempt to model the diurnal and seasonal dynamics of soil CO$_2$ exchange
(F$_S$) in a dryland ecosystem with a high plant-interspace heterogeneity. The modelling used an
integrated process-based approach, in which the CO$_2$ production, transport and surface exchanges (e.g.
biocrust photosynthesis, respiration and photodegradation) are considered simultaneously. The model
was parameterized and validated with multivariate data measured during year 2013-2014 in a semiarid
shrubland ecosystem in Yanchi, northwestern China.  We also investigated the sensitivity of simulated
F$_S$ to a set of stand-specific parameters and investigated the relative contribution of different flux
components. The model explained reasonably well the two-year dynamics of F$_S$ measured from a non-
crusted and two lichen-crusted plots. Simulations showed that the temporal pattern of F$_S$ could deviate
from that of the total CO$_2$ production from rooting-zone soil. Such deviations could be explained by
the variations of CO$_2$ dissolution and the CO$_2$ exchanges of biocrust during wetting-drying cycles, and
the root uptake and transport of dissolved CO$_2$. Moreover, the F$_S$ was spatially sensitive to the plant-
interspace differences and the variations in root biomass, soil organic matter and pH. These results
emphasized that, the processes beyond autotrophic and heterotrophic respirations and the
heterogeneities of soil at plant-interspace can strongly affect the F$_S$ dynamics and their climatic
sensitivities. Such variability should be carefully considered in extrapolation of findings from
chamber to ecosystem level and from seasonal to inter-annual scales. Based on this work, our model
can serve as a useful tool to simulate F$_S$ dynamics in dryland ecosystems.
Keyword: ecosystem modelling; heterogeneity; inorganic carbon; semiarid shrub ecosystem; biocrust





1.  Introduction
$CO_2$ exchange between soil and atmosphere constitutes a major C loss from terrestrial ecosystems
(Raich et al., 2002; Giardina et al., 2014). It also plays an important role in the feedbacks between
global carbon cycle and climate change (Rustad et al., 2000; Giardina et al., 2014; Karhu et al., 2014).
However, the contribution of soil $CO_2$ flux ($F_S$) in arid and semiarid (dryland) ecosystems to the
global C budget is less-studied (Castillo-Monroy et al., 2011; Gao et al., 2012; Jia et al., 2014),
although these areas cover over 40% of land surface and contribute notably to inter-annual variations
of terrestrial C sink (Poulter et al., 2014). The temperature dependency of biological $CO_2$ productions
(i.e. autotrophic respiration and heterotrophic respiration) serves a conventional basis for $F_S$ modelling
in many terrestrial ecosystems (Raich and Tufekciogul, 2000; Ryan, 2005; Song et al., 2015). Soil
$CO_2$ flux of dryland ecosystems is also widely interpreted using temperature-response functions
modified by other environmental constraints, e.g. soil water content, abundances of substrates and
microbial activities (Curiel Yuste et al., 2007; Wang et al., 2014a, 2014b, 2015).
Although empirical models may reproduce the dynamics of soil $CO_2$ flux in specified space-time,
their lack of mechanistic descriptions represents a major difficulty in extrapolation under changing
environmental conditions (Fan et al., 2015). Soil $CO_2$ flux is a "bulk" exchange that comprises two
main sets of processes, i.e. the $CO_2$ productions and transport (Fang and Moncrieff, 1999; Fan et al.,
2015). Hence, models considering only autotrophic and heterotrophic respiration often fail to account
for the observed $F_S$ dynamics (Austin and Vivanco, 2006). Gas transport processes are important
mechanisms regulating the magnitude and hysteretic feature of soil $CO_2$ efflux (Ma et al., 2013). A
substantial fraction of respired $CO_2$ may be transported to atmosphere via xylem, and can't be
measured by techniques like soil reparation chambers (Bloemen et al., 2013; 2016). During wet period,
soil $CO_2$ efflux could decrease significantly by water clogging of soil pores, which restricts the
diffusion of $O_2$ and $CO_2$ gases (Šimunek and Suarez, 1993; Fang and Moncrieff, 1999). In dryland
soils, the interactions between $CO_2$ transport and water cycle could also be intensive, due to the
commonly high salinity/alkalinity of soils. Large inorganic C fluxes can be driven solely by
dissolution and infiltration of $CO_2$ and carbonates (Buysse et al., 2013; Ma et al., 2013; Fa et al.,
2014). Such inorganic transport may not only contribute to the hourly or diurnal soil $CO_2$ efflux  (e.g.
Emmerich, 2003; Xie et al., 2009; Buysse et al., 2013), but also to the terrestrial $CO_2$ sinks at much
broader spatiotemporal scales (Schlesinger, 2009; Li et al., 2015).
Key processes contributing to $CO_2$ production in dryland soils also extend beyond autotrophic
respiration and heterotrophic respiration. Although biocrust organisms (lichens, mosses, bacteria,
fungi and microfauna) inhabit in the top few centimetres of the soil profile,  they constitute up to 70%
of biomes in the plant-interspace (Belnap, 2003). These communities are able to uptake C from the
atmosphere (Belnap, 2003; Castillo-Monroy et al., 2011; Maestre et al., 2013), leading to largely



greater concentration of organic matters in the crusted layer than the sub-soils (Ciais et al., 2013).
Although crust organisms could maintain inactive under stresses (e.g. drought, Green and Proctor,
2016), their photosynthetic potential could be large (Zaady et al., 2000; Lange, 2003), even
comparable to temperate forests with closed canopies (e.g. Zaady et al., 2000). The net C uptake by
biocrust is highly sensitive to stresses like droughts, thermal extremes and excessive ultraviolet
radiation (Pointing and Belnap, 2012). Such variations can readily alter the crusted soils between
considerable $CO_2$ sinks and sources within a few hours (e.g. Bowling et al., 2011; Feng et al., 2014).
In addition, the accumulation of debris from crust and canopy further fuels photodegradation, which
represents an important abiotic C loss in arid conditions beside the biotic decomposition (e.g. Austin
and Vivanco, 2006). Photodegradation  is likely to dominate the mineralization during dry daytime
period, when the radiation is strong and microbial activities are prohibited by low moisture content
and high temperature (e.g. Gliksman et al., 2016). On an annual basis, photodegradation could
consume more than 10% of soil organic matter (SOM) at surface (e.g. Austin and Vivanco, 2006;
Henry et al., 2008; Brandt et al., 2010). This could be the case even for the substrates (e.g. lignin) that
are difficult to degrade via biotic pathways (Henry et al., 2008).
The influences of multiple C processes (i.e. autotrophic and heterotrophic respiration, net C uptake
by biocrust, inorganic C fluxes and photodegradation) on soil $CO_2$ exchange are highly overlapped
and tightly related to the water-energy processes. In dryland ecosystems, patchy vegetation and large
fractions of interspace are common features (Domingo et al., 2000), and the water-thermal conditions
can vary considerably from plant cover to interspace even within a few meters (Rodríguez-Iturbe et al.,
2001; Caylor et al., 2008; Ma et al., 2011). The water-energy dynamics at the different surfaces are
linked by multiple advection processes both above- and below-ground (Gong et al., 2016). Due to the
complexity of water-energy processes, there may exist possibly high non-linearity of water-thermal
responses to the climatic variability (e.g. Phillips et al., 2011; Barron-Gafford et al., 2013). This will
also complicate the C responses and consequently affect the relationships between the $CO_2$ fluxes and
environmental controls (e.g. Jarvis et al., 2007; Song et al., 2015).
The global change is expected to increase annual mean air temperature considerably and change
precipitation regimes (Donat et al., 2016). Understanding the response of dryland ecosystems to such
changes requires mechanistic models that integrate the multiple biotic and abiotic mechanisms in soil
C cycling. So far, only a few models have coupled the biotic $CO_2$ productions with the transport of
gas and heat (Šimunek and Suarez, 1993; Fang and Moncrieff, 1999; Phillips et al., 2011; Ma et al.,
2013; Fan et al., 2015). Nevertheless, none of those models have described the heterogeneous water-
energy processes in soil-vegetation-atmosphere continuum (SPAC), or the unconventional C fluxes
such as net C uptake by biocrust and photodegradation despite the importance of these processes in
arid and semiarid environments. Models by Porada et al. (2013) and Kinast et al. (2016) represent the
few existing work in this sense. However, both models focus on the patterns at the regional-scale with



very simplified ecosystem processes and neglect stand-scale heterogeneities of water-energy budget,
and have not yet been validated by field measurements.
This study represents a first attempt to model the diurnal and seasonal dynamics of soil $CO_2$
exchange ($F_S$) in a dryland ecosystem with a high plant-interspace heterogeneity. The modelling used
an integrative process-based approach, in which the $CO_2$ production, transport and surface exchanges
(e.g. biocrust photosynthesis, respiration and photodegradation) are considered simultaneously. The
model was parameterized and validated by multi-variant data measured during year 2013-2014 in a
semiarid shrubland ecosystem in Yanchi, northwestern China. By employing the model, we also
investigated the sensitivity of simulated $F_S$ to a set of stand-specific parameters and investigated the
role of different flux components in regulating the $F_S$. The model development in this study is based a
water-energy modelling by Gong et al. (2016).

**2. Materials and methods**

**2.1 Outlines for the modelling**

The process-based modelling was based on multi-variate data measured during year 2013-2014 in
a semiarid shrubland ecosystem located at the southern edge of the Mu Us desert (37°42'1" N,
107°13'7" E, 1560 m above sea level), Ningxia, northwestern China (see Wang et al., 2014a, 2015).
The long-term mean temperature (1954–2004) is 8.1 °C, and the mean annual precipitation is 287 mm,
most of which falls from July to September (Jia et al., 2014). The radiation and evaporation demand
are high in this area, i.e. the annual incoming shortwave radiation is $1.4 \times 10^5$ J cm$^{-2}$ and the annual
potential evaporation is 2024 mm. The vegetation is dominated by scattered crowns of *Artemisia*
*ordosica* (Fig. 1a). The soil is highly alkaline (pH = 8.2). Biocrust (mainly lichens and algae) covers
about 40% of interspace soil. The thickness of the crust layer was 0.5 – 2.5 cm (Gong et al., 2016).
The modelled ecosystem was subtracted as replications of "representative land units" (RLU, Fig. 1;
Gong et al., 2016), which consist of the area covered by shrubs and the surrounding soil (interspace).
Vertically, the model simulates the C flows over the soil profile and the water-energy transport from
the lower boundary of rooting zone to a reference height in the boundary atmosphere. Horizontally,
the SPAC processes at plant cover and the surrounding interspace are differentiated but related via
advection and diffusion flows, as driven by the gradients of temperature, water potential and gas
concentrations. The mineralization, uptake and transport of soil C and N are further regulated by
water-energy conditions. Key processes and variables included in the FS modelling are shown in Fig.
1(c).
The model includes a set of sub-models, which describe: (i) $CO_2$ dissolution, transport and efflux;
(ii) Autotrophic and heterotrophic $CO_2$ productions in the soil profile; (iii) $CO_2$ uptake and emission
by biocrust; (iv) Surface energy balance and soil temperature profile; and (v) Soil hydrology and





water balance. These sub-models are linked by multiple feedbacks to represent the coupling of C,
water, vapor and energy transportations in the ecosystem. Sub-models (iv) - (v) have been developed
and described in details in our previous work (Gong et al. 2016), which focused on (i) introducing the
plant-interspace heterogeneity into water-energy modelling, and (ii) investigating the influences of
such heterogeneity on the ecosystem water-energy budgets for a dryland ecosystem. Gong et al. (2016)
also validated the model in regard to the diurnal to seasonal dynamics of radiation balance, surface
energy balance, soil temperature and moisture content in the footprint area of a eddy-covariance (EC)
site (details of measurement see Jia et al., 2014). In this work, we therefore focus on the development
of sub-models (i) – (iii) and their parameterization and validation by $F_S$ measurements, based on
automatic respiration chambers from crust-covered and non-crusted soils. Based on the validated
model, we also analyzed the model sensitivities to stand parameters and plant-interspace heterogenity
and investigated the relative contribution of different flux components to $F_S$.

**2.2 Modelling approaches**
**2.2.1 Submodel (i): $CO_2$ transport, dissolution and efflux**
For soil fraction $x$ (see Fig. 1b for RLU settings), $CO_2$ exchange ($F_S$, upward positive) was the sum
of $CO_2$ exchange by biocrust ($F_B$), photodegradation ($F_P$) and the total emission from soil under the
biocrust layer ($F_T$):
$$F_{S_x} = F_{B_x} + F_{T_x} + F_{P_x} \tag{1}$$
where $F_B$ is the net balance between biocrust photosynthesis ($P_B$) and respiration ($R_B$), and $F_B = P_B -$
$R_B$ (see Section 2.2.3). $F_T$ was modelled based on the mass-balance functions of PATCIS (Fang and
Moncrieff, 1999), which combined major transport processes in both gaseous and liquid phases. To
account for the plant-interspace heterogeneity, we expanded the original one-dimensional function to
the two-dimensional space. For soil layer ($x$, $i$) and time step $t$, the $CO_2$ concentration and C flows
were calculated as follows:
$$\frac{\partial c_{x,i}}{\partial t} = \frac{\partial}{\partial z}\left(F_{dg}^v + F_{ag}^v + F_{dw}^v + F_{aw}^v\right) + \frac{\partial}{\partial h}\left(F_{dg}^h + F_{ag}^h + F_{dw}^h + F_{aw}^h\right) + S_{x,i} \tag{2}$$
where superscripts $v$ and $h$ denote the vertical and horizontal directions, respectively (see also in Gong
et al., 2016); $C$ is the total $CO_2$ content; $F_{dg}$ and $F_{dw}$ are the $CO_2$ flows due to diffusion/dispersion via
the gaseous and liquid phases; $F_{ag}$ and $F_{aw}$ are the flows in gaseous and liquid phases due to gas
convection and water movement, and $S$ is the net $CO_2$ sink of the layer. The calculation schemes of
$F_{dg}$, $F_{dw}$, $F_{ag}$ and $F_{aw}$ have been described in detail by Fang and Moncrieff (1999). $F_T$ is the total
exchange of gaseous $CO_2$ between surface and topmost layer:
$$F_{T_x} = F_{dg_{x,1}}^v + F_{ag_{x,1}}^v + E_{x,1}^S C_{w_{x,1}} \tag{3}$$



where $E^S_{x,1}$ is the soil evaporation at section $x$ (see Eq. (17) in Gong et al., 2016); $C_w$ is the equivalent
$CO_2$ concentrations in the solution of the topmost soil. For layer $(x, i)$, $C_w$ is linked to the gaseous $CO_2$
concentrations ($C_g$):
$$C_{x,i} = Cg_{x,i}(V_{x,i} - \theta_{x,i}) + Cw_{x,i}\theta_{x,i} \qquad (4)$$
where $V$ is the total porosity; and $\theta$ is soil water content.
$C_g$ and $C_w$ were further related via the dissolution-dissociation balance of $CO_2$ in soil solution,
following Fang and Moncrieff (1999) and Ma et al (2013):
$$CO_2(g) + H_2O(l) \rightleftarrows H_2O(l) + CO_2(aq) \quad K_H = P_C/CO_2^{aq} \qquad (5)$$
$$CO_2(aq) + H_2O(l) \rightleftarrows H_2CO_3 \qquad K_0 = CO_2^{aq}/[H_2CO_3] \qquad (6)$$
$$H_2CO_3 \rightleftarrows [H^+] + [HCO_3^-] \qquad K_1 = [H^+][HCO_3^-]/[H_2CO_3] \qquad (7)$$
$$HCO_3^- \rightleftarrows [H^+] + [CO_3^{2-}] \qquad K_2 = [H^+][CO_3^{2-}]/[HCO_3^-] \qquad (8)$$
where $P_C$ is the partial pressure of $CO_2$ in pore air; $K_H$ is Henry's Law constant; $K_0$, $K_1$ and $K_2$ are
equilibrium coefficients of dissolution, the first- and the second-order dissociation reaction for
carbonic acid, respectively (for details see Fang and Moncrieff, 1999). The equilibrium $[H^+]$ was
determined by the soil pH and the coefficients $K_H$, $K_0$, $K_1$ and $K_2$, which were functions of soil
temperature in each soil layer (Fang and Moncrieff, 1999). $Cw$ was calculated as the sum of $CO_2^{aq}$,
$H_2CO_3$, $HCO_3^-$ and $CO_3^{2-}$.

**2.2.2 Submodel (ii): autotrophic and heterotrophic $CO_2$ productions along the soil profile**
For soil layer $(x, i)$, $S_{x,i}$ (Eq. 2) was calculated as the sum of autothrophic and heterotrophic $CO_2$
productions, and the dissolved $CO_2$ removed with the water uptaken by roots:
$$S_{x,i} = Rs_{x,i} + Ra_{x,i} - E_{x,i}Cw_{x,i} \qquad (9)$$
where $E$ is the transpirative uptake of water (Gong et al., 2016); $Rs$ is the $CO_2$ production by
heterotrophic SOM decomposition; $Ra$ is the autotrophic respiration of the rhizosphere, which
comprises maintenance respiration ($Rm$) and growth respiration ($Rg$):
$$Ra_{x,i} = Rm_{x,i} + Rg_{x,i} \qquad (10)$$
To simulate $Rs$, we simplified the pool-type model of Gong et al (2013, 2014), which was
originated from Smith et al (2010) for simulating coupled C and N cycling in organic soils. SOM pool
in each soil layer was divided into debris ($M_{deb}$, i.e. litters from roots and biocrust), microbes ($M_{mic}$)
and humus ($M_{hum}$), which are different in biochemical recalcitrance and N content. During decaying,
mineralized masses transfer from $M_{deb}$ and $M_{mic}$ to more resistant form (i.e. $M_{hum}$), leading to a
decrease in lability (e.g. Li et al., 1992). The mineralization of organic C followed first-order kinetics



and was constrained by multiple environmental multipliers, including temperature, water content and
oxygen content (Šimunek and Suarez, 1993; Fang and Moncrieff, 1999):
$$m_{x,i}^r = M_{x,i}^r k_r f(Ts_{x,i}) f(\theta_{x,i}) f(O_{x,i}) dt \qquad (11)$$
where superscript $r$ denotes the type of SOM pool ($r$=1 for $M_{deb}$, $r$=2 for $M_{mic}$, and $r$=3 for $M_{hum}$,); $m$ is
mineralized SOM during time step $dt$; $k$ is the decomposition constant; $dt$ is time step; $f(Ts_{x,i})$, $f(\theta_{x,i})$
and $f(O_{x,i})$ are multiplier terms regarding the temperature, water content and oxygen restrictions,
respectively. $f(O_{x,i})$ was calculated following Šimunek and Suarez (1993). $f(Ts_{x,i})$ and $f(\theta_{x,i})$ were
reparameterized with respect to the site-specific conditions of plants and soil (see Section 2.3.4). The
$CO_2$ production from mineralization was further regulated by the N-starvation of microbes following
Smith et al. (2010):
$$Rs_{x,i} = r_E m_{x,i}^r \qquad (12)$$
where $r_E$ is the gas production rate ($r_E \in [0, 1]$), and (1- $r_E$) is the proportion of organic matters passed
to the downstream SOM pools. The evolution of each SOM pool was calculated as below:
$$M_{x,i}^r = (1 - r_E) m_{x,i}^{r-1} - m_{x,i}^r + A_{x,i}^r dt \qquad (13)$$
where $A$ is the SOM input rate ($A$=0 for $M_{mic}$ and $M_{hum}$); superscript $r$-1 denotes the source SOM pools.
$Rm_{x,i}$ was calculated in a similar way to $Rs_{x,i}$ (e.g. Chen et al., 1999; Fang and Moncrieff,1999). $Rg_{x,i}$
was calculated as a fraction of photosynthetic assimilates, following Chen et al. (1999):
$$Rm_{x,i} = M_{x,i}^R k_R f(Ts_{x,i}) f(\theta_{x,i}) f(O_{x,i}) dt \qquad (14)$$
$$Rg_{x,i} = k_g P_g fr_{x,i} \qquad (15)$$
where $M^R$ is the root biomass; $k_R$ is the specific respiration rate of roots; $k_g$ is the fraction of
photosynthetic assimilate consumed by growth respiration; $fr_{x,i}$ is the mass fraction of roots in soil
layer ($x$, $i$). $P_g$ is the photosynthesis rate of plants. $P_g$ was estimated using a modified Farquahar's leaf
biochemical model (see Chen et al., 1999). This model simulates photosynthesis based on
biochemical parameters (i.e., the maximum carboxylation velocity, $V_{max}$, and maximum rate of
electron transport, $J_{max}$), foliage temperature ($Tc$) and stomatal conductance ($gs$). The values of $V_{max}$
and $J_{max}$ were obtained from in situ measurements from the site (Jia et al., unpublished). $Tc$ and $gs$
were given in the energy balance sub-model, which was detailed in Gong et al. (2016).
N content bonded in SOM mineralized and was added to soil layers simultaneously with decaying.
The abundance of mineral N (i.e. $NH_4^+$ and $NO_3^-$) regulates the growth of microbial biomass and $r_E$
following Smith et al. (2010) and Gong et al. (2014). Key processes governing the dynamics of
mineral N pools included nitrification-denitrification (Smith et al., 2010), solvent transport with water
flows (Gong et al, 2014) and the N uptake by root system. However, the plant growth was not
modelled in this work and therefore, $N_{upt}$ was calculated using the steady-state model of Yanai (1994),





based on the transpiration rate, surface area of fine roots and the diffusion of solvents from pore space
to root surface:
$$N_{upt} = 2\pi r_o L \alpha C_o dt \qquad (16)$$
where $r_o$ is the fine root diameter; $L$ is the root length, and $2\pi r_o L$ is the surface area of fine roots; $\alpha$ is
the nutrient absorbing power, which denotes the saturation degree of solute uptake system ($\alpha \in [0,1]$);
$C_o$ is the concentration of solvents at the root surface, and is a function of bulk concentration of
mineral N ($N_{min}$), inward radial velocity of water at the root surface ($v_o = E/(2\pi r_o L)$) and saturation
absorbing power $\alpha$. Further details for calculations of $\alpha$ and $C_o$ can be found in work of Yanai (1994).

### 247 2.2.3 Submodel (iii): CO₂ exchange of biocrust and photodegradation

Biocrusts are vertically layered systems that comprise topcrust (or, bio-rich layer) and underlying
subcrust (or, bio-poor layer), which are different in microstructure, microbial communities and C
functioning (Garcia-Pichel et al., 2016; Raanan et al., 2016). Topcrust is usually few-millimetre thick,
which allows the penetration of light and the development of photosynthetic microbes (Garcia-Pichel
et al., 2016). On the other hand, the subcrust has little photosynthetic-activity. We here focused
mainly on describing the C exchanges in the topcrust, but assumed the C processes in the subcrust
were similar to those in the underneath soil. We developed the following functions to describe the C
fixation and mass balance in the topcrust,
$$F_{Ct} = P_{Ct} - R_{Ct} \qquad (17)$$
where $P_{Ct}$ is the bulk photosynthesis rate; and $R_{Ct}$ is the bulk respiration rate. $P_C$ and $R_C$ were further
modelled as follows:
$$P_{Ct} = \frac{\alpha_C A_{PAR} P_{Cm}}{\alpha_C A_{PAR} + P_{Cm}} \qquad (18)$$
$$R_{Ct} = M_{Ct} k_{cr} f_{RC}(T_{Ct}) f_{RC}(\theta_{Ct}) \qquad (19)$$
where $\alpha_C$ is the apparent quantum yield, $P_{Cm}$ is the maximal rate of photosynthesis, and was a function
of the moisture content ($\theta_{Ct}$) and temperature ($T_{Ct}$) in topcrust; $A_{PAR}$ is the photosynthetically active
radiation (PAR); $M_{Ct}$ is the total C in the SOM of topcrust; $k_{cr}$ is the respiration coefficient; $f(\theta_{Ct})$ and
$f(T_{Ct})$ are water and temperature multipliers. Here, we assumed no photosynthesis in subcrust. The
heterotrophic respiration ($R_{Cs}$) was calculated as was done for soil respiration (Eq. (11)) based on the
C storages ($M^r_{x,1}$) and temperature and moisture content of crust layer (i.e. $Ts_{x,1}$ and $\theta_{x,1}$; see Eq. (29)
and Eq. (14) in Gong et al., 2016).
To consider different C losses and exchanges, and to calculate the C balance in topcrust and
subcrust, respectively, we considered the following matters. $R_{Ct}$ includes the respirations from both





autotrophic ($M_{CA}$) and heterotrophic ($M_{CH}$) pools. When autotrophic organisms die, SOMs pass from $M_{CA}$ to $M_{CH}$ and influence the turnover processes. A variety of topcrust organisms can reach into subcrust (e.g. through rhizines, Aguilar et al., 2009) and export litters there. When the surface is gradually covered by deposits, topcrust organisms tend to move upward and recolonize at the new surface (e.g. Garcia-Pichel and Pringault, 2001; Jia et al., 2008), leaving old materials buried into the subcrust (Felde et al., 2014). On the other hand, the debris left to soil surface are exposed to photodegradation. Based on above, the C balance in topcrust and subcrust was calculated as following, assuming the partitioning of respiration between autotrophic and heterotrophic pools was proportional to their fractions:

$$M_{Ct} = M_{CA} + M_{CH} \tag{20}$$

$$\frac{dM_{CA}}{dt} = P_{Ct} - R_{Ct}\frac{M_{CA}}{M_{Ct}} - k_m M_{CA} - k_b M_{CA} \tag{21}$$

$$\frac{dM_{CH}}{dt} = k_m M_{CA} - R_{Ct}\frac{M_{CH}}{M_{Ct}} - k_b M_{CH} - F_P \tag{22}$$

$$\frac{dM_{Cs}}{dt} = k_b M_{Ct} - R_{Cs} \tag{23}$$

where $k_m$ is the rate of C transfer (e.g. mortality) from autotrophic pool to heterotrophic pool; $k_b$ is the rate of C transfer (e.g. burying) from topcrust to subcrust; $F_P$ is the loss of SOM due to photodegradation.

Photodegradation tends to decrease surface litter masses in a near linear fashion with the time of exposure (Austin and Vivanco 2006; Vanderbilt et al., 2008). Considering the diurnal and seasonal variations of radiation, $F_P$ was calculated as a function of surface SOM mass and solar radiation:

$$F_{P_x} = M_{surf} k_p Rad_x \tag{24}$$

where $Rad_x$ is the incident shortwave radiation at surface $x$ (Gong et al., 2016); $M_{surf}$ is the surface litter mass; and $k_p$ is the photodegradation coefficient.

### 2.3 Model parameterization

### 2.3.1 Measurements of micrometeorology and soil CO$_2$ efflux

Meteorological variables were measured every 10 seconds and aggregated to half-hourly resolution during 2013-2014. The factors measured included the incoming and outgoing irradiances (PAR-LITE, Kipp and Zonen, the Netherlands), PAR (PAR-LITE, Kipp and Zonen, the Netherlands), air temperature and relative humidity (HMP155A, Vaisala, Finland). Rainfall was measured with a tipping bucket rain gauge (TE525WS, Campbell Scientific Inc., USA) mounted at a nearby site (1 km



away, see Wang et al., 2014a). The seasonal trends of the measured $Ta$ and $P$ can be found in Jia et al.
(2016). No surface runoffs were observed at the site, indicating the horizontal redistribution of rainfall
was mainly through subsurface flows.
Continuous measurements of $F_S$ were conducted using an automated soil respiration system (model
LI-8100A fitted with a LI-8150 multiplexer, LI-COR, Nebraska, USA). The system was on a fixed
sand dune of typical size (Wang et al., 2014a), which was located about 1.5 km south from the EC
tower described in Gong et al (2016). Three collars (20.3 cm in diameter and 10 cm in height, of
which 7 cm inserted into the soil) were installed on average at 3m spacing in March 2012. One collar
(C1) was set on a bare soil microsite with no presence of biocrust. Two other chambers (C2 and C3)
were set on lichen-crusted soils. $F_S$ was measured hourly from C1 and C2 by opaque chambers,
whereas by transparent chamber from C3 to include the photosynthesis and photodegradation. Litters
from the shrub canopies were cleared from the collars during weekly maintenance. Hourly $Ts$ and $\theta$ at
10 cm depth were measured outside each chamber using the 8150–203 soil temperature sensor and
ECH2O soil moisture sensor (LI-COR, Nebraska, USA), respectively. Root biomass was sampled
near each collar (within 0.5 m) in July 2012, using a soil corer (5 cm in diameter) to a depth of 25 cm.
The samples were mixed and sieved sequentially through 1, 0.5 and 0.25 mm meshes, and the living
roots were picked by hands. The comparison of the three micro-sites is shown in Table 1. Methods
used in data processing and quality control have been described earlier in details (see Wang et al.,
2014a, 2015). The quality control led to gaps of 10 - 13% in the $F_s$ dataset.

### 2.3.2 Parameterization of vegetation and soil texture

The parameterization schemes supporting the simulations of energy balance and soil hydrology in
sub-model (i) - (v) have been described previously in detail by Gong et al. (2016). As the water-
energy budget is sensitive to vegetation (i.e. canopy size, density and leaf area) and soil hydraulic
properties (see Gong et al., 2016), we hereby revalued these parameters for the $F_S$ site. Measurements
based on four 5m×5m plots showed that the crown diameter $D$ ($86 \pm 40$ cm) and height $H$ ($47\pm 20$ cm)
at this site were similar to those measured from the eddy-covariance (EC) footprint by Gong et al.
(2016). However, the shrub density was 50% greater, leading to higher shrub coverage (42%), shorter
spacing distance $L$ (40.2 cm) and greater foliage area. On the other hand, the subsoil at the $F_S$ site is
sandy and much coarser than that at the EC footprint. Therefore, we collected 12 soil cores from 10
cm depth, and measured saturated water content ($\theta_{sat}$), bulk density and residual water content ($\theta_r$)
from each sample. Then, the samples were saturated, and covered and drained by gravity. We
measured the water content after 2-hour and 24-hour draining, which roughly represented the matrix
capillary water content (10 kPa) and field capacity (33 kPa) (Armer, 2011). The shape parameters $n$
and $\alpha_h$ (see Eq. (26) in Gong et al. 2016) for the water-retention function were estimated from these
values (Table 2).




### 2.3.3 Parameterization of soil C and N pools

The sizes and quality of soil C pools were parameterized based on a set of previous studies. The
total SOC in the root-zone soil (i.e. 60cm depth, bulk density of 1.6 g cm$^{-3}$) was set to 1200 g m$^{-2}$,
based on the values reported from previous studies in Yanchi area (e.g. Qi et al., 2002; Chen and
Duan, 2009; Zhang and Hou, 2012; Liu et al., 2015; Lai et al., 2015). The mass fraction of resistant
SOM pool ($M_{hum}$) was set to 40 - 50 % of total SOM, following work by Lai et al. (2015). The vertical
distribution of the SOM pools was described following Shi et al. (2013). At the ecosystem level, the
total root biomass was calculated as proportional to the aboveground biomass (Xiao et al., 2005),
which linearly related to the crown projection area ($0.5\pi D^2$, Zhang et al., 2008). The vertical profile of
root biomass was parameterized following Li and Xiao (2007), and the root biomass was set to
decrease with the distance from the center of a shrub crown (Zhang et al., 2008). The N content was
parameterized following the measurement of Wang et al. (2015).
Based on the above settings, the specific decomposition rate of debris was estimated from the
litterbag experiment of Lai et al. (2015), which showed a 16% decrease in the mass of fine-root litters
during a 7-month period of year 2013 at the Yanchi site. The photodegradation coefficient ($k_p$) was
calculated from the mass-loss rate reported by Austin and Vivanco (2006). $M_{surf}$ was set to 33% of
$M_{CH}$ in topcrust, assuming the depth of light penetration was about 2 mm and C concentration was
homogeneous in topcrust. The surface litter from canopy was not considered in this modelling, as the
plant litters were cleaned from the collars during weekly maintenance. The specific respiration rate of
roots ($k_R$), however, could be much greater during vegetative growing stage than other periods, e.g. at
the defoliation stage (Fu et al., 2002; Wang et al., 2015). Here we linked $k_R$ to the development of
foliage in modelling using the approach of Curiel Yuste et al. (2004):
$$k_R = k_{R0}(1 + n_R L_l/L_{max})$$    (25)
where $k_{R0}$ is the "base" respiration rate (Table 2); $L_l$ is the green leaf area, which is a function of Julian
day (Gong et al., 2016); $L_{max}$ is the maximum $L_l$; $n_R$ is the maximum percentage of variability and is
set to 100%.

### 2.3.4 Parameterization of the water-thermal sensitivity of soil CO$_2$ productions

Based on the empirical study of Wang et al. (2014a), the steady-state sensitivity of CO$_2$ productions
to soil temperature and water content (i.e., $f(Ts)f(\theta)$, Eq. (11)) can be described as a logistic-power
function:
$$f(Ts)f(\theta) = f(Ts, \theta) = \{1 + \exp[a(b - Ts)]\}^{-1}(\theta/\theta_{sat})^c$$    (26)



where *a*, *b* and *c* are empirical parameters. This function represents the long-term water-thermal
sensitivity of $CO_2$ productions over the growing seasonal, yielding an apparent temperature sensitivity
$Q_{10}$ of 1.5 for the emitted $CO_2$ (Wang et al. 2014a). However, this could underestimate the short-term
sensitivities of $CO_2$ productions. The apparent $Q_{10}$ could be much greater at the diurnal level than at
the seasonal level (Wang et al., 2014a). In this work, we firstly calculated the "base" sensitivity using
the long-term scheme (Eq. 26) with 1-day moving average of water-thermal conditions. Then the
deviation of hourly sensitivity from "base" condition was adjusted by the short-term $Q_{10}$:
$$f(Ts)f(\theta) = f(Ts_{short}, \theta_{short}) + [f(Ts, \theta) - f(Ts_{short}, \theta_{short})]Q_{10}^{(Ts - Ts_{short})/10} \quad (27)$$
$$Q_{10} = \max[Q_{10}(Ts_{short}), Q_{10}(Ts_{short})] \quad (28)$$
$$Q_{10}(Ts_{short}) = -0.42\, Ts_{short} + 12.4 \quad (29)$$
$$Q_{10}(Ts_{short}) = 18010\, \theta_{short}^{3.721} + 1.604 \quad (30)$$
where $Ts_{short}$ and $\theta_{short}$ are the 1-day moving averages of $Ts$ and $\theta$, respectively; $Q_{10}(Ts)$ and $Q_{10}(\theta)$
are the adjustment functions for short-term apparent $Q_{10}$, regarding the short-term $Ts$ and $\theta$.
Further non-linearity of soil respiration responses refers to the rain-pulse effect (or the "Birch
effect", Jarvis et al. 2007), that respiration pulses triggered by rewetting can be orders-of-
magnitudegreater than the value before rain event (Xu et al., 2004; Sponseller, 2007; Cable et al.,
2013). Such response could be very rapid (e.g. within 1 hour to 1 day, Rey et al. 2005) and sensitive
to even minor rainfalls.  It also seems that the size and duration of a respiration pulse not only depend
on the precipitation size, but also on the moisture conditions prior to the rainfall (Xu et al., 2004; Rey
et al., 2005; Evans and Wallenstein, 2011). In this work, we multiplied a simple rain-pulse coefficient
($f_{pulse}$) to Eq. (26):
$$f_{pulse} = max[1, (\theta/\theta_{72h})^{n_p}] \quad (31)$$
where is the 3-day moving average of soil moisture content; $n_p$ is a shape parameter and was set to 2
in this study. $\theta_{72h}$ is the 72-hour moving average of $\theta$.

**2.3.5 Parameterization of biocrust photosynthesis and respiration**
In sub-model (iii), Equations (17) - (19) were parameterized based on the experiment of Feng et al.
(2014). In the experiment, 50 lichen (topcrust) samples of 0.5-0.7 cm thickness (100% coverage,
average C content of 1048 umol C $cm^{-3}$) were collected from a 20 m × 20 m area. The samples were
wetted and incubated under controlled $T_{Ct}$ (i.e. 35°C, 27°C, 20°C, 15°C, and 10°C). These samples
were divided into two groups to measure the net primary productivity (NPP) and dark respiration (Rd)
separately. Gas exchanges and light response curve for each crust sample were measured using LI-
6400 infrared gas analyzer equipped with an LI-6400-17 chamber and an LI-6400-18 light source (LI-



COR, Lincoln, NE, USA). Measurements were taken at ambient $CO_2$ values of $385 \pm 35$ ppm.
Saturated topcrust samples were placed in a round tray and moved to the chamber. $CO_2$ exchange was
measured during the drying of samples, until the $CO_2$ flux diminished. During drying, $\theta_{Ct}$ was
measured every 20 min. For more details see Feng et al. (2014).
Fitting the measured Rd to $T_{Ct}$ and $\theta_{Ct}$ (see Fig. 2a) obtained the multipliers in Eq. (19) as following:
$$f_{RC}(T_{Ct})f_{RC}(\theta_{Ct}) = Q_{Ct}^{\frac{(T_{Ct}-20)}{10}}(a_{RC} + b_{RC}\theta_{Ct} + c_{RC}\,\theta_{Ct}^2) \qquad (32)$$
where $Q_{Ct}$, $a_{RC}$, $b_{RC}$, $c_{RC}$ are the fitted shape parameters (Table 2).
The parameterized Eq. (19) was then used to simulate the Rd for the NP samples, based on the
correspondent $T_{Ct}$ and $\theta_{Ct}$ of each measurement. $P_{Cm}$ was determined by subtracting the simulated
respiration rate from the NP measured under light-saturated conditions. Then $P_{Cm}$ was fitted to $T_{Ct}$
and $\theta_{Ct}$ as following (Fig. 2b):
$P_{Cm} = f_{Pt}(T_{Ct})\,f_{Pw}(\theta_{Ct})$
$$= (a_{Pt} + b_{Pt}T_{Ct} + c_{Pt}\,T_{Ct}^2 + d_{Pt}\,T_{Ct}^3)(-a_{Pw} + b_{Pw}\theta_{Ct} - c_{Pw}\,\theta_{Ct}^2 + d_{Pw}\,T_{Ct}^3) \quad (33)$$
where $a_{Pt}$, $b_{Pt}$, $c_{Pt}$, $d_{Pt}$, $a_{Pw}$, $b_{Pw}$, $c_{Pw}$, $d_{Pw}$ are fitted shape parameters (Table 2).
It should be addressed that $T_{Ct}$ and $\theta_{Ct}$ could change more rapidly than the mean conditions of the
crust (i.e. $Ts_{x,1}$ and $\theta_{x,1}$). In this work, $T_{Ct}$ was calculated from the surface temperature ($T_x$, see Eq. (13)
in Gong et al., 2016) and $Ts_{x,1}$ by linear interpolation. The calculation of $\theta_{Ct}$, on the other hand,
depended on the drying-rewetting cycle. During drying phases, $\theta_{Ct}$ was interpolated linearly from $\theta_{x,1}$
and surface moisture content ($\theta_x$); whereas during wetting phases, the mass balance of water input $P$
and evaporation loss ($E^s_{x,1}$, see Eq. (17) in Gong et al., 2016) was considered:
$$T_{Ct} = \frac{T_x Z_{Ct} + Ts_{x,1} Zs_{x,1}}{Z_{Ct} + Zs_{x,1}} \qquad (34)$$
$$\theta_{Ct} = max\left[\frac{\theta_x Z_{Ct} + \theta_{x,1} Zs_{x,1}}{Z_{Ct} + Zs_{x,1}}, \theta_{Ct} + \frac{P - E^S_{x,1}}{Z_{Ct}}\right] \qquad (35)$$
where $Zs_{x,1}$ is the thickness of the biocrust; and $Z_{Ct}$ is the thickness of the topcrust. $\theta_x$ was calculated
from the surface humidity and the water retention of the crust layer, using Eq. (25) – (26) by Gong et
al. (2016).

### 2.3.6 Calculation of litter input to soil and SOC transport in biocrust

The litter falls added to each soil layer ($A^1_{x,i}$, Eq. (13)) were linked to the mortality of roots, which
was calculated following Asaeda and Karunaratne (2000).



$$A_{x,i}^1 = k_{mo} Q_{mo}^{Ts_{x,i}-20} M_{x,i}^R \qquad (36)$$
where $k_{mo}$ is the optimal mortality rate at 20°C; $Q_{mo}$ is the temperature sensitivity parameter (Asaeda
and Karunaratne, 2000). Similarly, we attributed the C transport rate ($A_{Cm}$) from $M_{CA}$ to $M_{CH}$ mainly to
the mortality of autotrophic organisms. We assumed that most mortality of crust organisms occurred
during abrupt changes in wetness, as microbial communities may adapt slow moisture changes or
remain inactive during drought (e.g. Roberson and Firestone, 1992; Reed et al., 2012; Coe et al., 2012;
Garcia-Pichel et al., 2013; Maestre et al., 2013). Here, we introduced a water-content multiplier,
$f_m(\theta_{Ct})$, to describe the impact of abrupt $\theta_{Ct}$ changes on $k_m$:
$$A_{Cm} = k_{mc} Q_{mo}^{T_{Ct}-20} f_m(\theta_{Ct}) M_{CA} \qquad (37)$$
$$f_m(\theta_{Ct}) = \max[0.01, 1 - \min(\theta_{Ct}, \theta_{Ct7})/\max(\theta_{Ct}, \theta_{Ct7})] \qquad (38)$$
where $k_{mc}$ is the optimal mortality rate at 20°C; $Q_{mo}$ is the temperature sensitivity parameter (Asaeda
and Karunaratne, 2000); $\theta_{Ct7}$ is the forward 7-day moving average of $\theta_{Ct}$.
C transport from topcrust to subcrust was calculated as driven mainly by the sand deposition and
burying of topcrust SOM. Assuming the C content in topcrust was homogeneous and the thickness $Z_{Ct}$
was near-constant, the transport rate ($k_b$) was then proportional to the sand deposition rate:
$$k_b = \frac{k_{sand}}{\rho_{bulk}} \frac{1}{Z_{Ct}} \qquad (39)$$
where $\rho_{bulk}$ is the bulk density of soil; $k_{sand}$ is the sand deposition rate in Yanchi area, which is a
function of wind velocity (Li and Shirato, 2003):

**2.4 Model validation and sensitivity analyses**
**2.4.1 Boundary conditions and initial values used in model simulations**
In the model simulations, soil depth was set to 67.5 cm to cover the rooting zone (Gong et al.,
2016), including the crust layer (2.5 cm) and sandy subsoil (65 cm, stratified into 5 cm layers). Water
content measured at 70 cm depth was used as the lower boundary condition for hydrological
simulation (Jia et al., 2014). The calculation of soil temperature extended to 170 cm depth with no-
flow boundary, regarding the probably of strong heat exchange at the lower boundary of rooting zone
(Gong et al., 2016). Zero-flow condition was set for the lower boundary of $CO_2$ and $O_2$ gases, whereas
dissolved $CO_2$ was able to leech with seepage water. Based on presumed similarity of RLU structures,
we assumed no-flux conditions for transports of water, heat, solvents and gases at outer boundary. In
the simulation, we assumed instant gas transport via topcrust, whereas considered the $CO_2$ released by
subcrust ($R_{Cs}$) was subject to the dissolving-transport processes. In this work, we aggregated the C




processes in subcrust with those in soil profile and set $F_B = F_{Ct}$ (Eq. (2)). The initial ratio of $M_{CA} : M_{CH}$
was set to 2:3. The C concentration of organic matters was set to 50%.

464       The model simulation employed half-hourly meteorological factors including the incoming

shortwave radiation, incoming longwave radiation, PAR, $Ta$, relative humidity, wind speed and
precipitation. The initial temperatures and soil moisture content for each soil layer were initialized
following the work by Gong et al. (2016). Surface $CO_2$ concentration was set to 400 ppm. The initial
gaseous $CO_2$ concentration was set to increase linearly with depth (5 ppm $cm^{-1}$). The initial $CO_2$
concentration in liquid form was then calculated based on Eq. (4) – Eq. (8). The initial content of
mineral N content was set to 40 mg/g, which was within the range of the field observations. The two-
dimensional transpirations of water, energy and gases along the soil profile were solved numerically
using the Predict–Evaluate–Correct–Evaluate (PECE) method. In order to avoid undesired numerical
oscillations, the transport of water, energy and gases were calculated at 5-min sub steps.

### 2.4.2 Model validation

First, we validated the modelling of soil temperature and moisture content for the $F_S$ site (Test 0). The
simulated hourly soil temperature and moisture content at 10 cm depth were compared to the
measured values for each collar. The validation was based on the same meteorological data as used by
Gong et al. (2016), who validated the model in regard to the diurnal to seasonal dynamics of radiation
balance, surface energy balance, soil temperature and moisture content at the EC site.

481       The validity of the modelled $F_S$ was examined using three separate tests. In Test 1, modelled $F_S$ was

validated for non-crusted soils. In this case, $F_T$ in Eq. (1) was the only term affecting $F_S$ ($F_B = 0$ and $F_P$
= 0), and the crust influences on C-water exchanges were excluded. The biocrust-relatedprocesses
were considered in Test 2 and Test 3. Test 2 considered the dark respiration of biocrust ($R_{Ct}$), and set
$F_B = R_{Ct}$ and $F_P = 0$. Test 3 considered all the flux components ($F_T$, $F_P$ and $F_B$). In these tests, different
values of root biomass were assigned to the model regarding the different collar conditions (Table 1).
In Test 1 – Test 3, half-hourly $F_S$ were simulated and averaged to hourly values, and compared to the
those measured from the collar C1 – C3, respectively. Linear regressions were used to compare the
modelled and measured values. The biases ($\zeta$) of the simulated values were calculated by subtracting
the measured values from the modelled ones. Gap values in the measurements were omitted in the
validation and the bias analyses.

### 2.4.3 Sensitivity analyses

By employing the validated model, we studied the sensitivity of simulated soil $CO_2$ efflux to a set of
stand-specific parameters and compared the C fluxes at plant-cover and interspace soils (Test 4). This
was done to find out how different flux components (i.e. $P_{Ct}$, $R_{Ct}$, $F_P$, $F_T$, $Ra$ and $Rs$) to the soil $CO_2$





efflux. These simulations were based on model settings for C3 and climatic variables of year 2013-
2014. For the comparison purpose, same settings for soil C storages (650 gC m$^{-2}$), root biomass (200 g
m$^{-2}$) and biocrust were employed at both under-canopy and interspace areas.
Our previous work has already studied the sensitivities of modelled soil temperature and moisture
content to the variations in soil texture, water retention properties, vegetation parameters and plant-
interspace heterogeneities Gong et al. (2016). In this study, we further tested the sensitivity of the
modelled $F_S$ and componental fluxes to the changes in a number of stand-specific variables and
parameters (Table 4).
We also investigated the model sensitivities for several newly defined parameters (i.e. $n_R$, $n_p$ and $f_m$).
It was also studied the sensitivities of modelled C fluxes to the changes in soil temperature and
moisture content, which could be biased regarding the heterogeneities of soil texture, hydraulic
properties and vegetation covers (see e.g. Ma et al., 2011; Gong et al., 2016). We further on analysed
the model sensitivity to a set of biogeochemical parameters, including root biomass, total SOC ($M_{tot}$ =
$M_{deb}$ + $M_{mic}$ + $M_{hum}$) in soil and topcrust ($M_{Ct}$), soil N content ($N_{tot}$), ratio of $M_{CA}$ : $M_{CH}$, the
decomposability of debris ($k_l$), the rates of litter productions from roots ($k_{mo}$) and topcrust ($k_{mc}$) and
soil pH. Simulated $F_S$ and componental fluxes at the interspace were also compared to the values
under no-change conditions.

**3. Results**
**3.1 Model validity**
Figure 3 shows the modelled hourly $Ts$ and $\theta$ at 10 cm depth with the mean values measured from
the $F_S$ site during year 2013. Based on the site-specific vegetation and soil texture parameters, our
model explained 97% of the variations in the measured hourly $Ts$. The model underestimated the
temperature mainly in summer time (i.e. day 150-250, Fig. 3a). The underestimation was most
pronouncing around the noontime in the diurnal cycle. The measured $\theta$ at 10 cm depth was much
lower at the $F_S$ site than that shown by Gong et al. (2016) for the EC site (Fig. 3b). Such a difference
was in line with the coarser texture of the soil at the $F_S$ site. The model underestimated mainly the soil
water content during the freezing season (Fig. 3b). During the ice-free period, it explained 83% of the
variations in the measured mean water contents at 10 cm depth.
The measured $F_S$ showed large diurnal and seasonal variations regardless the existence of crust
cover (Fig. 4 and Fig. 5). Rain events clearly influenced the hourly $F_S$ measured from the non-crusted
surface C1 (Fig. 4a). The $F_S$ dropped significantly from the pre-rainfall level even to near-zero, but
rebound rapidly and peaked after rain stopped. Our modelling reasonably reproduced the diurnal and
seasonal fluctuations of $F_S$. The model explained 87 and 83% of the variations in the hourly $F_S$
measured on the non-crusted surface in year 2013 and 2014, respectively (Fig. 4a). The model mainly



underestimated the daytime $F_S$ during the freezing seasons. During the ice-free periods, the model
overestimated the efflux in early springs. The biases of modelling largely showed a diurnal pattern
(Fig. 4c), that $F_S$ was underestimated in noon hours (i.e. from 10 a.m. to 3 p.m.) but overestimated in
the afternoon and evening. At the daily level, our model explained 94% of the variations in measured
daily efflux during the two-year period (Fig. 4c).

537        Comparing to the non-crusted surface (C1), the simulated $F_S$ for the crusted surfaces (C2 and C3)

exhibited greater deviations from the observations. At the hourly scale, our model explained 75 %
(year 2013) and 68 % (year 2014) of the variations of measured $F_S$ from C2 (Fig. 5a), and 68 % (year
2013) and 61 % (year 2014) of the variations of measured $F_S$ from C3 (Fig. 5b). For the two-year
period, the root-mean-square errors (RMSE) of the modelled hourly $F_S$ were 0.25 umol m$^{-2}$ s$^{-1}$ and
0.35 umol m$^{-2}$ s$^{-1}$ for C2 and C3, respectively. The biases of the simulated $F_S$ for C2 and C3 showed
similar diurnal pattern as compared to C1, and the magnitudes of biases ($|\zeta|$) were greater during the
rainfall period (i.e. from the start of raining to 24 hours after end of rainfall) than the inter-rainfall
period (Fig. 6). Nevertheless, at the daily scale, the model explained 91% (C2, Fig. 5c) and 86% (C3,
Fig. 5d) of the variations in the measured $F_S$ during the two-year period. There were no significant
systematic deviations between the measured and the modelled daily values, as indicated by regression
slopes close to 1 and intercepts close to 0 (Figs. 4 and 5).

**3.2 Model sensitivity**
*Relative contribution of component fluxes to $F_S$*

552        If root biomass and SOC were homogeneous at plant cover and interspace, the C loss at interspace

was faster than under-canopy on an annual basis (Test 4, Table 3). In both areas, $Rs$ was a major
contributor to the total $CO_2$ produced in root-zone soil and $F_T$ dominated the effluxes ($F_S$) during the
two-year period. The simulated NPP of topcrust were 18.2 gC m$^{-2}$ year$^{-1}$ and 31.1 gC m$^{-2}$ year$^{-1}$ at
under-canopy and interspace, respectively. At hourly scale, the net C uptake by topcrust could be
comparable to $F_T$ after rewetting (Fig. 5A). However, at annual scale, the C losses via respiration and
photodegradation accounted for 90% of the GPP, leading to a near-zero contribution of topcrust to $F_S$
during the two year period (i.e. $< 5$ gC m$^{-2}$ year$^{-1}$).

560        Test 4 further showed mismatched trends of $F_T$ and the root-zone $CO_2$ production ($R_P$). The annual

$F_T$ was 17 and 15% smaller than $R_P$ at under-canopy and interspace, respectively. Such a gap was
mainly due to the root uptake and transport of dissolved $CO_2$ (i.e. 36 gC m$^{-2}$ year$^{-1}$) whereas the $CO_2$
loss via seepages or pore-mediated horizontal flows were limited (i.e. 7.4 gC m$^{-2}$ year$^{-1}$). Moreover,
the temporal patterns of $F_T$ and $R_P$ were largely inconsistent with respect to the wetting-drying cycles
(Fig. 7a). Comparing to $R_P$, the responses of $F_T$ to rainfall were largely lagged and smoothed (Fig. 7b
− 7d), disregard the size of rain events. $R_P$ increased rapidly following the rewetting of soil. On the
other hand, $F_T$ firstly depressed during rainfall then increased after rain ceased. In all the examples, $F_T$



exceeded $R_P$ within 48 hours after the ending of rain events. At the annual level, the total $R_P$ was
larger during wetting period (i.e. raining days plus 1 day after rainfall) than the rest days of the year
(i.e. drying period), whereas the total $F_T$ was greater during the drying period (Fig. 5e).

*Sensitivity of modelled $F_S$ to site-specific parameters*

In general, the modelled $F_S$ and the component fluxes were more sensitive to ± 2 ºC in $Ts$ or ± 10%
in $\theta$, comparing to the effects of ± 10% or ± 20% in the other parameters (Table 4). Varying $\theta$ by 10%
produced greater impacts on the simulated $R_P$ and crust-related fluxes (i.e. $P_{Ct}$, $R_{Ct}$ and $F_P$), as
compared to changing $Ts$ by ± 2 ºC. Increasing $\theta$ by 10% enhanced the simulated $P_{Ct}$ and $NP_{Ct}$ of the
crust by 41 and 28%, and doubled the annual C sequestration by topcrust. However, such changes in
crust C uptake had minor contribution to $F_S$, as it amounted for only 2.0% of the total efflux.
Adjustment of the newly introduced parameters $N_P$ and $f_m$ by ± 20% produced limited influences on
the modelled $F_S$ and the componential fluxes (Table 4). The model was also robust to the adjustment
of several crust-related parameters, i.e. $k_{mc}$, $M_{Ct}$ and $M_{CA}$: $M_{CH}$. Comparing to ± 10% in root biomass,
varying $N_R$ by 20% led to similar responses in the simulated $F_A$ but much weaker responses in $F_T$ and
$F_S$. A 10% variation in root biomass changed the annual $F_T$ and $F_s$  by about 7%, and such effects
were 100% greater than that of 10% changes in the total SOC ($M_{tot}$).

**4. Discussions**

**4.1 Model performance**

*Validity of the $F_S$ modelling at non-crusted soil*

Our model reasonably well captured the measured dynamics of soil temperature during year 2013,
based on revaluing the vegetation and soil hydraulic parameters for the $F_S$ site. The model mainly
underestimated the midday temperatures at the collars. In addition, the model underestimated the soil
moisture content during the winter freezing period, probably because the impacts of solvents on the
thermal conductivity and freezing point of soil water (Viterbo et al., 1999) were not included (see
Gong et al., 2016). This could have contributed to the biases in the simulated $Ts$ and $F_S$ for freezing
period. However, the influences of such biases on annual $F_S$ was marginal, due to the very low
emissions during the winter period (see also Liu et al., 2016). During ice-free season, the simulated
soil water content agreed well with the measured means, and the biases in the modelled temperature
and moisture content were less than the spatial variations observed in the area (e.g. Wang et al., 2015).
Therefore, our model could be able to reproduce near-realistic trends for the water-energy conditions
at the site. Based on the modelled water-energy dynamics, the model well described the seasonal
variations of $F_S$ measured from the non-crusted soil (Test 1) during a two-year period. The model was
able to capture the strong variability of hourly/daily $F_S$ in wetting-drying cycles, and the performance
was generally better than those based on empirical methods (e.g. Wang et al., 2014a, 2014b). In this





sense, we assume that our model have included the main mechanisms controlling the $F_S$ dynamics in the non-crusted soil system.

The uncertainties in the modelling, however, may exist in several aspects. *Firstly*, the RLU was a statistical simplification to the target ecosystem at footprint scale (Gong et al., 2016), and may not fully capture the spatially explicitly of soil environment and biogeochemistry at the scale of FS measurements. For example, the model assumed Poisson probability of mutual shading (Bégué et al., 1994), and the probability of shading increased continuously with solar zenith (Gong et al., 2016). However, for explicit space-time, the chance of being sunlit or shaded is more likely to be binary, which  possibly explain the underestimation of net radiation (Gong et al., 2016) and the collar temperature around midday, affecting the simulated $F_S$ (see Fig. 3b). Moreover, field observations had considerable spatial variations in soil temperature, water content and biogeochemistry (e.g. pH, litter quality and root biomass) within a distance of 3-5 meters. Such variations could well exceed a magnitude of 10 %, and even over 100 %  (e.g. Zhang et al., 2008; Feng et al., 2013; Wang et al., 2015). Therefore, the variation of $F_S$ driven by the spatiality of soil factors could be greater than the responses to ±2 °C in soil temperature or ±10 % in soil water content. Therefore, future modelling may need to consider spatially explicit settings, in order to further minimize the gaps between model settings and the reality, and to improve the reliability of simulated $F_S$  responses to changing climatic conditions.

*Secondly*, the uncertainties in inorganic C processes could strongly affect the accuracy of modelling, as indicated by the high sensitivity of simulated $F_S$ to soil pH. In this model, the calculation of $CO_2$ transport was based on gaseous and liquid phases, whereas the solid phase were not involved. This is likely to underestimate the dissolved $CO_2$ and its fluctuations, regarding the high lime content (2300–5400 kg ha$^{-1}$) in the soil (Feng et al., 2013; Wang et al., 2015). Based on soil samples of similar lime content (2700 kg ha$^{-1}$), Buysee et al. (2013) showed that neglecting the inorganic C exchanges of the $CaCO_3$-$H_2O$-$CO_2$ system could underestimate $F_S$  during the heating phase of a day, but overestimate $F_S$ during the cooling phase. Such phenomenon was very similar to the diurnal pattern of biases found in our modelling (Fig. 3d). Therefore, further improvement on the modelling may need to consider the solid phase as well. However, the effect of solids could be complicated, as it may differ by the mineral type and degree of exposure to pore air and liquid (Buysee et al., 2013; Fa et al., 2014). Liu et al. (2015) suggested that a considerable fraction of $CO_2$ absorbed by minerals could even be stored for prolonged period with high biochemical stability. In this sense, the functioning of the $CO_2$ buffering system may require site-specific parameterizations, in order to improve the model performance at hourly level.

*In addition*, the current model still lacked mechanistic descriptions on the growth of plants. Comparing to many other ecosystems, drylands feature high root-shoot ratio (Jackson et al., 1996) but low SOC storages. Changes in plant physiology and growth can readily influence root metabolisms and labile SOC pools, thus modify the climatic sensitivities of $F_S$  (Wang et al., 2015). On the other




hand, the parameterization of soil respiration employed constant scheme throughout the years.
However, the large fluctuation of diurnal and seasonal temperature may drive the microbial
communities to shift between warm-adapted to cold-adapted (Van Gestel et al., 2013), which largely
enhances winter respiration and its sensitivity to freeze-thaw cycles (Van Gestel et al., 2013; Liu et al.,
2016). Both the biotic controls are mixed with the legacy effects of the climatic variability over
annual and inter-annual courses (Sala et al., 2012; Jia et al., 2016; Shen et al., 2016), and could affect
the C-water simulations cumulatively through the feedbacks between biomass accumulation and soil
biogeochemistry (Bradford et al., 2016). This may explain the decrease of model validity from year
2013 to 2014 (Fig. 3, Fig. 4). Therefore, the dynamics of plants and microbial communities are
required in future modelling, in order to improve the $F_S$ simulations regarding inter-annual and long-
term periods.

*Validity of the $F_S$ modelling at lichen-crusted soils*
Comparing to the non-crusted soil, the adding of biocrust increased the model uncertainty
particularly at the hourly scale (e.g. Fig. 4). The biases of modelled $F_S$ for the lichen-crusted soil
mainly exist at hourly level after rewetting. Nevertheless, the model reasonably explained the seasonal
dynamics of the measured $F_S$ during 2013-2014, and the RMSE of the modelled hourly $F_S$ was one-
order smaller than the seasonal variation of the measured values. This indicated that the model could
serve a quantitative tool to simulate seasonal and inter-annual $F_S$ from the lichen-crusted soils in
dryland ecosystems similar to the site studied here.
The uncertainty of the $F_S$ modelling for the lichen-crusted soils was partly due to the simulated
subsoil processes, as indicated by the similar diurnal patterns of biases across Test 1-3 (Fig. 3, Fig. 5).
Other sources of uncertainty may be due to the simulated crust processes, and especially due to the
biases of the estimated water content in topcrust (Table 4). The water content in the very thin layer of
topcrust can be highly dynamical during wetting-drying cycles. Hence, it is challenging to track the
photosynthesis or respiration peaks based on the hourly simulations of micrometeorology and soil
hydrology. Moreover, nocturnal water inputs (e.g. dewfalls) often occur at stable atmospheric
conditions. These rewetting events are important to the metabolisms of crust organisms (e.g. Liu et al.,
2006). However, they are hard to be quantified precisely by EC measurement, or models derived from
EC data. These uncertainties further affect together with the biases in the modelling of water-energy
balances (Gong et al., 2016). Due to the lack of data on the crust moisture dynamics, it is still difficult
to analyse the extent to which such uncertainties could have influenced the model validity.
The challenges to reproduce realistic trends of biocrust C uptake may also relate to the presumed
homogeneity of structures of the crust layers and the consistency of C-water processes throughout the
simulation. In reality, there may not be clear boundaries between topcrust and subcrust, and even
topcrust itself may contain significant variations in microstructure and communities even within one



centimetre (Williams et al., 2012; Raanan et al., 2016). Furthermore, the C sequestration of biocrust
not only relies on instantaneous environmental factors such as radiation, temperature and water
content (Feng et al., 2014), but also depends on the dynamics of microbial communities and their
interactions (Belnap, 2003; Pointing and Belnap, 2012; Maestre et al., 2015). The changes in
microbial communities, on the one hand, can influence the C functioning of biocrusts directly (Feng et
al., 2014; Maestre et al., 2015). On the other hand, such changes affect surface albedo (Chamizo et al.,
2012), nocturnal water inputs (Liu et al., 2006), soil aggregate structure and pore forming (Williams et
al., 2012; Felde et al., 2014), which ultimately feedback to microniches and water-gas fluxes (Garcia-
Pichel et al., 2016). So far, many questions remain unanswered about the mechanisms that control the
colonization, adaption and succession of microbial communities and the structure-function of biocrust
(Pointing and Belnap, 2012). Further knowledge on these mechanisms will be helpful to improve the
modelling of crust C functioning in response to climate change and extreme climatic events.

### 4.2 Effects of plant-interspace heterogeneity and soil C processes on soil $CO_2$ effluxes

Clumped distributions of foliage and biomass are critical features for the adaptation and functioning
of vegetation in arid and semiarid environments. Previous studies have mainly emphasized the effects
of shrub pattern on ecohydrology (e.g. Rongo et al., 2006; Gong et al., 2016) and enrichment of fine
sediments and nutrient, known as "resource island" effects (Reynolds et al., 1999; Rietkerk et al.,
2004). Our simulations showed that the presence of shrub canopy could also influence the soil C
exchanges. Comparing to the interspace, the presence of shrub cover reduced the simulated $F_S$ by 13%
annually (Test 4). As the soil SOC and root biomass were set to be the same at under canopy and
interspace in the simulation, such a decrease in $F_S$ was probably due to the cooling effect of canopy
(Gong et al., 2016) on soil. Such an effect was close to the modelled responses of $F_S$ to ± 2 °C in soil
temperature or ± 10% in soil water content. As the density of roots and litter production are
commonly larger under canopy than interspace (e.g. Zhang et al., 2008), the lower respiration rate
under canopy tends to facilitate the accumulation of biomass and organic matters in under-canopy
soils and feedback to functioning of "resource islands" during prolonged periods. In this context, the
different C functioning at plant cover and interspace shall not be neglected in studies on the climatic
sensitivity of C dynamics in dryland ecosystems.
Our modelling provide a way to separate the multiple soil C processes and investigate their roles in
regulating $F_S$ dynamics in dryland ecosystems. So far, efforts to quantify the soil C loss in terrestrial
ecosystems often consider soil C efflux as a synonym of respired $CO_2$. However, based on this work
cautions must be taken when extrapolating the $F_S$ responses from the chamber to ecosystem scale and
from short-term to long-term periods. Our simulations reckoned that a considerable fraction of $CO_2$
produced could be removed by root uptake and leave the volume measured by the respiration chamber.
Bloemen et al. (2016) showed that the $CO_2$ concentration in root xylem could be higher than in soil



solutions. This implies that such a "missing source" might be even greater than the model estimation,
although our knowledge is still limited about the efficiency of the removal and the diffusion/release of
$CO_2$ during the transport (Bloemen et al., 2016). On the other hand, the soil processes other than
autotrophic and heterotrophic respiration could significantly modify the $F_S$ responses to climatic
variability. Our simulation highlighted decoupled $CO_2$ productions and emissions during the wetting-
drying cycle, as regulated by the $CO_2$ transport in soil profile. The simulated $CO_2$ productions in soil
profile were much greater than effluxes during rain pulses (e.g. Fig. 7). This indicated that, the limited
FS during rewetting was mainly due to the increased dissolution of $CO_2$, rather than the reduced
respiration rates by low $O_2$ supply (e.g. Fang and Moncrieff, 1999). This finding is further supported
by the measurement of Maier et al. (2011), which showed that 40% of the respired $CO_2$ could be
stored temporally in soil pore-space after rainfalls. The dissolved $CO_2$ then released gradually with the
evaporation of pore water, leading to lagged responses of efflux as compared to respirations.
Regarding that a major fraction of $CO_2$ was produced during the wetting periods (Fig. 5e), such a
lagging effect should be carefully examined when analysing the climatic sensitivity of $F_S$ .
Accounting for the effects of biocrust organisms is also important to accurately estimate C
exchanges in dryland ecosystems (Maestre and Cortina, 2003; Castillo-Monroy et al., 2011). Existing
studies on the C functioning of biocrust have mainly focused on measuring the net C exchanges under
laboratory conditions or from field (e.g., Maestre and Cortina 2003; Wilske et al., 2008; Bowling et
al., 2011). However, the contributions of biocrusts as C sink or source remain largely unknown
(Castillo-Monroy et al., 2011), due to the difficulty to separate the $CO_2$ exchanges of crust organisms
from the background respirations (Castillo-Monroy et al., 2011; Sancho et al., 2016). In this sense, use
of our mechanistic model may provide more insights on the roles of biocrusts in soil C cycling and
effluxes. our simulation study showed that the C exchanges of biocrust was largely masked by the
background effluxes from root-zone soil. The C uptake by biocrust turned the soil from a net $CO_2$
source to sink during large rewetting events only, when the background emission was restricted. After
rain stopped, the sinks diminished quickly (e.g. within 1 day, Fig. 5b – 5c), not only due to the
decreased photosynthesis with drying, but also the increased $CO_2$ emission from the soil underneath.
Based on the climatic variables of a two-year period (Test 4), the simulated NPP of the topcrust was
31 g C m$^{-2}$ year$^{-1}$ at the interspace conditions. Considering a 30% coverage of lichens over the
sampling area (Feng et al., 2014), the interspace-level NPP was 9.3 g C m$^{-2}$ year$^{-1}$. This value was
largely greater than the lab-based estimation for the site (Feng et al., 2014), but it was in range of the
values reported from several other dryland ecosystems (i.e. 5–3 - 29 g C m$^{-2}$ year$^{-1}$, Sancho et al.,
744   2016).

Our simulations also suggested that photodegradation might offset about 48 % of the $CO_2$
photosynthesized by biocrust and reduced the net sequestration to about 5 gC m$^{-2}$ year$^{-1}$. It could
explain the much higher $F_S$  measured from the transparent chamber (C3) than the opaque chamber
(C2) during dry daytime periods (e.g. Fig. 8). It should be also noticed that the litter from shrub



canopy was not included in the measurement nor modelling. Also, the interactions between photodegradation and biotic decaying were not considered either. Therefore, the contribution of photodegradation to soil C balance could be greater than our estimation at the ecosystem level (see e.g. Gliksman et al., 2016). Future studies are therefore required to clarify the role of photodegradation in regulating the C turnover and sequestration of biocrusts in arid and semiarid ecosystems.

## 5. Conclusions

This work represents a first attempt to integrate the $CO_2$ production, transport and surface exchanges (e.g. biocrust photosynthesis, respiration and photodegradation) in $F_S$ modelling for dryland ecosystems of high plant-interspace heterogeneities. Our model simulated reasonably well the $F_S$ dynamics measured from non-crusted and lichen-crusted soil collars during year 2013-2014, although introducing the gas exchanges of lichen crust into modelling decreased the model performance at the hourly scale. Our model could thus be used to simulate the seasonal and annual $F_S$ in dryland soils similar to our site. However, further development of model may still be required on several aspects, e.g. by including. i) the spatial-explicit schemes for surface conditions and soil biogeochemistry; ii) influences of lime and solids on $CO_2$ transport; iii) growth dynamics of plants; iv) high-resolution dynamics of surface water-thermal conditions and v) the dynamics of microstructure and microbial communities of biocrusts.

Our modelling work also highlighted that, the plant-interspace heterogeneity and complexity of soil C processes could affect largely the soil $CO_2$ efflux. The presence of plant cover tended to decrease the $CO_2$ production from root-zone soil probably due to the cooling effect of canopy. Moreover, the transport processes of inorganic C and the metabolisms of biocrusts strongly modified the $CO_2$ efflux, and these influences are closely linked to soil hydrology. The $CO_2$ emission from root-zone soil also delayed by increased $CO_2$ dissolution after rewetting. In addition, an ineligible fraction of respired $CO_2$ could be removed via lateral flows and root uptakes, and be "missing" from volumes under respiration chambers. During rewetting, the lichen-crusted soil could shift from net $CO_2$ source to sink, due to the activated photosynthesis of lichens and the restricted $CO_2$ emissions from subsoil. Whereas after rain events, the NPP of lichens could be easily masked by the background C emissions from the soil profile. Based on our modelling, the annual NPP was 9.3 gC m$^{-2}$ by topcrust at interspace. However, the net C sequestration by topcrust could be marginal, if the photodegradation is accounted.

To conclude, our work suggests that the complexity and plant-interspace heterogeneities of soil C processes affect largely the soil $CO_2$ efflux dynamics and their climatic sensitivities, which should be carefully considered in extrapolation of findings from chamber to ecosystem level and from seasonal to inter-annual scales. Our model can also serve as a useful tool to simulate the soil $CO_2$ efflux dynamics in dryland ecosystems.



Acknowledgement
This modelling work was carried out under the Finnish-Chinese research collaboration project
EXTREME (2013-2016) by the University of Eastern Finland and Beijing Forestry University. The
instrumentation and field measurements utilized in this work were supported by National Natural
Science Foundation of China (NSFC) (Proj. No. 31361130340, No. 31670710 and 31670708) and
Beijing Forestry University. The modelling work of EXTREME project was supported by the
Academy of Finland (proj. no. 14921) and the University of Eastern Finland. Thanks to Peng Liu,
Huishu Shi, Yuming Zhang, Sijing Li, Zhihao Chen, Siling Tang, Yajuan Wu and Yuan Li for
assistance on the field measurements and instrumentation maintenance.

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





Tables

Table 1. Configuration of soil collars used in this study

| Collars | C1 | C2 | C3 |
|---|---|---|---|
| Surface type | Non-crusted | Lichen-crusted | Lichen-crusted |
| Chamber type | Opaque | Opaque | Transparent |
| Root biomass (g m$^{-3}$) | 420 | 106 | 92 |
| Gap of data (%) | 12.9 | 10.5 | 9.85 |
| Annual C efflux (gC m$^{-2}$) [a] | 259 | 194 | 192 |

[a] The values were calculated from the measured hourly FS data excluding data gaps.


Table 2. Parameters for soil water retention and C turnover

| Parameter | Equation | Unit | Value |
|---|---|---|---|
| $\alpha_h$ | - [a] | - | 0.0355 [b] |
| $n$ | - [a] | - | 1.5215 [b] |
| $k_1$ | (11) | g g$^{-1}$ day$^{-1}$ | 0.01 [c] |
| $k_2$ | (11) | g g$^{-1}$ day$^{-1}$ | 0.08 [d] |
| $k_3$ | (11) | g g$^{-1}$ day$^{-1}$ | 0.001 [d] |
| $k_g$ | (15) | g g$^{-1}$ | 0.15 [e] |
| $k_{cr}$ | (19) | g g$^{-1}$ s$^{-1}$ | 0.0014 [f] |
| $k_{R0}$ | (25) | g g$^{-1}$ day$^{-1}$ | 0.002 [e] |
| $a$ | (26) | - | 0.1 [g] |
| $b$ | (26) | - | 24 [g] |
| $c$ | (26) | - | 0.89 [g] |
| $Q_{Ct}$ | (32) | - | 1.585 [f] |
| $a_{RC}$ | (32) | - | -0.0525 [f] |
| $b_{RC}$ | (32) | - | 2.602 [f] |
| $c_{RC}$ | (32) | - | -1.653 [f] |
| $a_{Pt}$ | (33) | - | 0.9837 [f] |
| $b_{Pt}$ | (33) | - | -0.1385 [f] |
| $c_{Pt}$ | (33) | - | 0.0095 [f] |
| $d_{Pt}$ | (33) | - | -1.6318E-4 [f] |
| $a_{Pw}$ | (33) | - | -0.3501 [f] |
| $b_{Pw}$ | (33) | - | 5.5884 [f] |
| $c_{Pw}$ | (33) | - | -7.1783 [f] |
| $d_{Pw}$ | (33) | - | 2.6837 [f] |

[a] See Eq. (26) in Gong et al. (2016). Sources of parameter values: [b] This study, see section 2.3.2; [c] Lai et al.
(2015); [d] Gong et al. (2014); [e] Chen et al. (1999); [f] This study, see section 2.3.4 and Fig. 2; [g] Wang et al., 2014a.






Table 3. Simulated annual $F_S$ (gC m$^{-2}$ year$^{-1}$) and its componential fluxes (gC m$^{-2}$ year$^{-1}$) at plant cover and
interspace

| Surface type | $F_S$ | $F_T$ | $R_P$ [a] | $Ra$ | $P_{Ct}$ | $F_{Ct}$ | $F_P$ | $F_{Cnet}$ [b] |
|---|---|---|---|---|---|---|---|---|
| Interspace | 244 | 249 | 295 | 113 | 54.6 | 31.1 | 26.1 | 5.0 |
| Plant covered | 214 | 218 | 263 | 108 | 36.3 | 18.2 | 14.6 | 3.6 |

[a] $R_P$ is the total respired $CO_2$ in root-zone soil and is the sum of autotrophic respiration ($Ra = \sum_i Ra_i$, see Eq. (10))
and heterotrophic respiration($Rs=\sum_i Rs_i$, see Eq. (12)); [b] $F_{Cnet}$ is the net CO2 exchanges of topcrust and $F_{Cnet}$=
$F_{Ct}$ - $F_P$, see Eq. (17) – Eq. (18).
Table 4. Sensitivity of simulated $F_S$ and its componential fluxes to changes in parameter values

| Change of parameter | $F_S$ | $F_T$ | $R_P$ | $Ra$ | $P_{Ct}$ | $F_{Ct}$ | $F_P$ | $F_{Cnet}$ |
|---|---|---|---|---|---|---|---|---|
| $n_R$ +20 % | +3.3 | +3.2 | +2.7 | +7.9 | / | / | / | / |
| $n_R$ -20 % | -2.9 | -2.8 | -3.4 | -8.8 | / | / | / | / |
| $n_P$ +20 % | +1.6 | +1.6 | +1.0 | / | / | / | / | / |
| $n_P$ -20 % | / | / | -1.4 | / | / | / | / | / |
| $f_m$ +20 % | / | / | / | / | +2.9 | +3.8 | +3.4 | +6.0 |
| $f_m$ -20 % | / | / | / | / | +1.2 | / | -5.7 | +30 |
| $Ts$ +2 ℃ | +9.5 [a] | +9.6 | +7.1 | +11 | +4.9 | +3.9 | +1.5 | +16 |
| $Ts$ -2 ℃ | -9.0 | -9.2 | -8.1 | -11 | -1.3 | -2.9 | / [b] | -20 |
| $\theta$ +10 % | +3.6 | +5.6 | +7.5 | +14 | +41 | +28 | +14 | +102 |
| $\theta$ -10 % | -5.0 | -5.6 | -8.1 | -14 | -16 | -13 | -8.4 | -34 |
| $M_{tot}$ +10 % | +2.9 | +2.8 | +2.0 | / | / | / | / | / |
| $M_{tot}$ -10 % | -2.5 | -2.4 | -3.1 | / | / | / | / | / |
| $M^R$ +10 % | +7.0 | +6.8 | +6.8 | +8.8 | / | / | / | / |
| $M^R$ -10 % | -7.0 | -6.8 | -7.1 | -8.9 | / | / | / | / |
| $N_{tot}$ +10 % | / | / | / | / | / | / | / | / |
| $N_{tot}$ -10 % | / | / | / | / | / | / | / | / |
| $k_1$ +10 % | +2.9 | +2.8 | +2.4 | / | / | / | / | / |
| $k_1$ -10 % | -2.5 | -2.4 | -3.1 | / | / | / | / | / |
| $k_{mo}$ +10 % | +4.1 | +4.0 | +3.4 | / | / | / | / | / |
| $k_{mo}$ -10 % | -3.3 | -3.2 | -3.7 | / | / | / | / | / |
| $k_{mc}$ +10 % | / | / | / | / | / | / | +1.5 | -8.0 |
| $k_{mc}$ -10 % | / | / | / | / | / | / | -2.3 | +8.0 |
| $M_{Ct}$ +10 % | / | / | / | / | / | / | / | / |
| $M_{Ct}$ -10 % | / | / | / | / | / | / | / | / |
| $M_{CA}:M_{CH}$ +10 % | / | / | / | / | / | / | / | / |
| $M_{CA}:M_{CH}$ -10 % | / | / | / | / | / | / | / | / |
| pH +5 % | -8.6 | -8.4 | / | / | / | / | / | / |
| pH -5 % | +7.0 | +6.8 | / | / | / | / | / | / |

[a] Percentage (%) of changes in the C flux after manipulation of parameter values, as compared to the "base"
conditions (i.e. Test 4; see Table 3). All the tests were based on the interspace conditions. [b] The change in the
simulated C flux was smaller than 1 % .





Figures

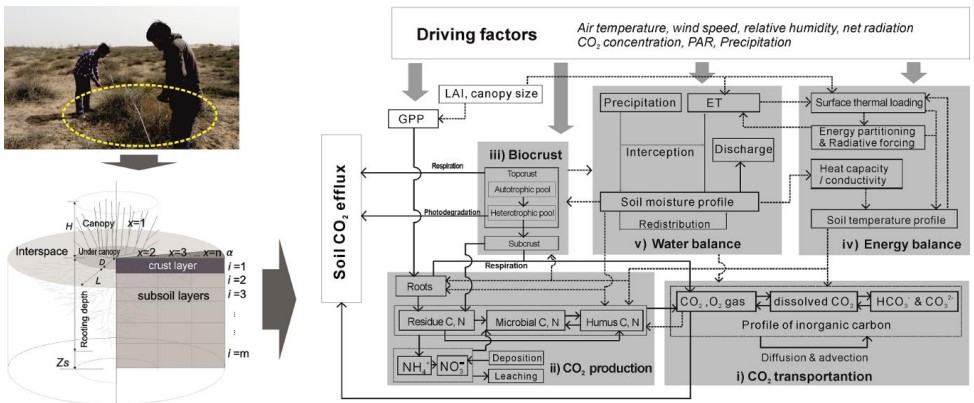


Figure 1. Layout for the setting of representative land unit (RLU, as adopted from Gong et al., 2016)
and conceptual framework of process-based modelling

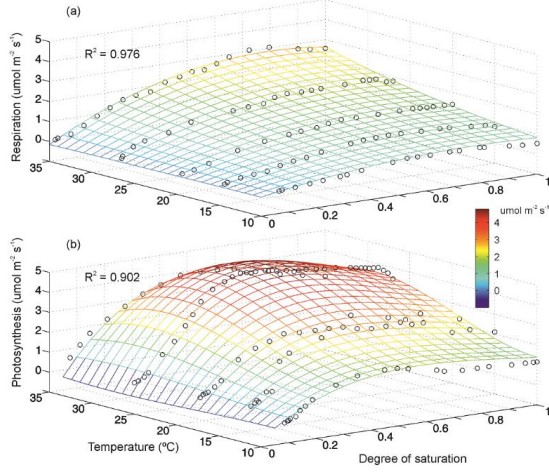


Figure 2. Measured and fitted bulk respiration (a) and photosynthesis (b) of the lichen topcrust as
functions of temperature and water content.





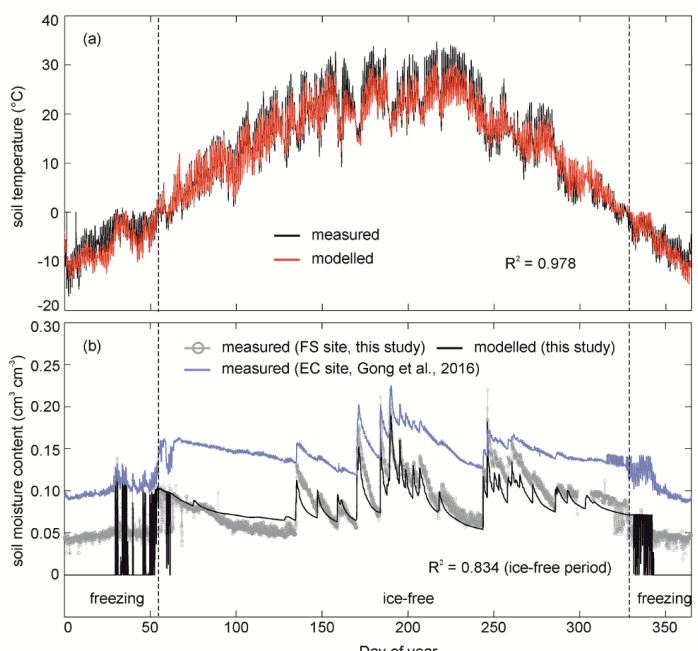


Figure 3. Measured and modelled soil temperature (a) and soil moisture content (b) at 10 cm depth for
F$_S$ site, and as compared to the EC site in year 2013 by Gong et al. (2016).


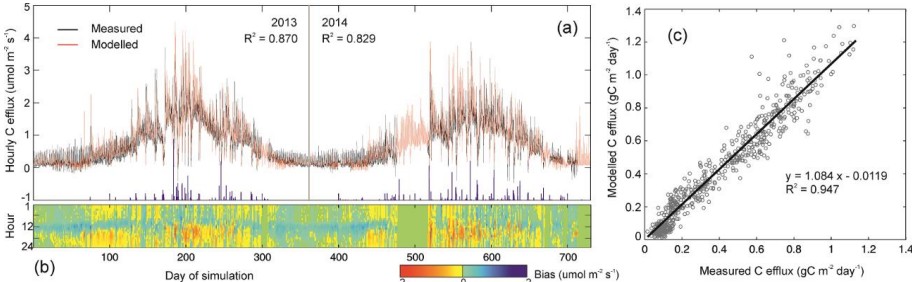


Figure 4. Measured and modelled hourly F$_S$ for non-crusted soil (a), the temporal pattern of the bias of
simulated hourly F$_S$ (b) and the comparison of measured and modelled daily F$_S$ (c) during 2013-2014.





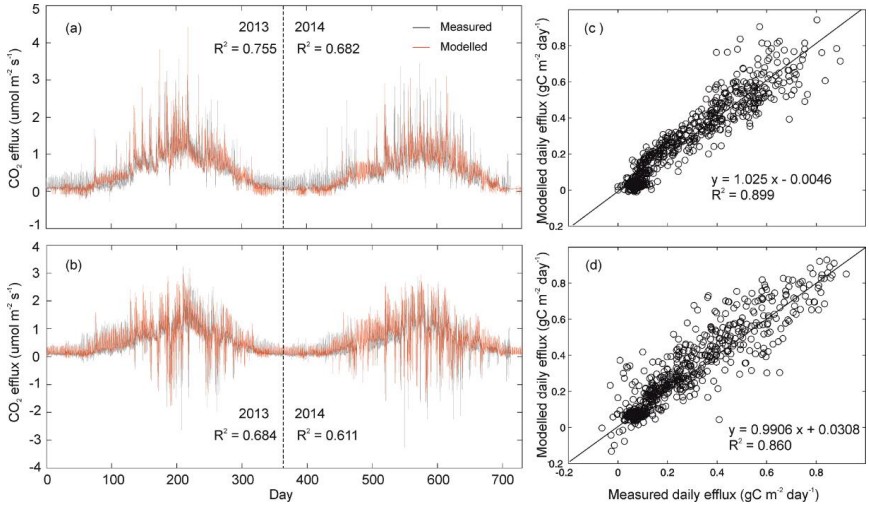

Figure 5. Measured and modelled $F_S$ of lichen-crusted soils for opaque (a, c) and transparent

chambers (b, d) at hourly (a, b) and daily (c, d) scales during 2013-2014.

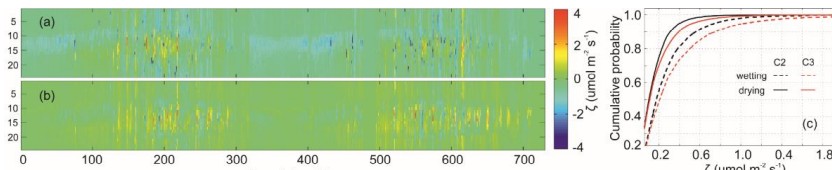

Figure 6. Diurnal patterns of biases in the simulated hourly $F_S$ for lichen-crusted soils using opaque (a)

and transparent chambers (b), and the cumulative probability of the biases during wetting and drying

periods (c) during 2013-2014. The wetting period included the raining days and a 1-day period after

each rainfall. The drying period included the rest time of the years other than the wetting period.

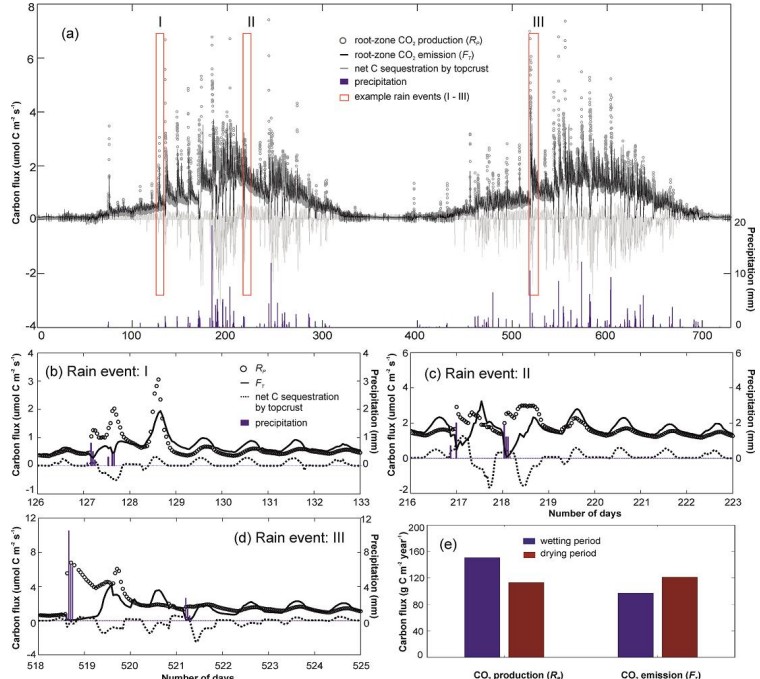

1116

Figure 7. Simulated $F_S$ and $CO_2$ exchanges by biocrust and root-zone soil (a), the simulated $CO_2$

fluxes before and after example rain events of 2.3 mm (b), 7.6 mm (c) and 12.8 mm (d) sizes, and the

comparison of $F_T$ and $R_R$ during wetting and drying periods during 2013-2014. The wetting period

included the raining days and a 1-day period after each rainfall. The drying period included the rest

time of the years other than the wetting period.

1122

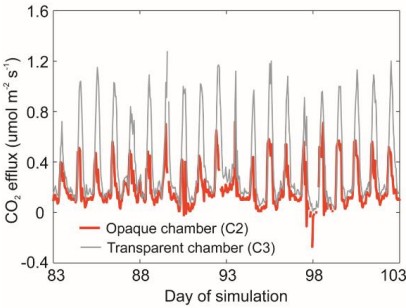

1123

Figure 8. Comparison of the measured $F_S$ from lichen-crusted surfaces using opaque and transparent

chambers during a dry period (day 83-103) in spring 2013.