# Peer review of "high plant-interspace heterogeneity Jinnan Gong1, Ben Wang1,2,, Xin Jia1,2,, Wei Feng2, Tianshan Zha2, Seppo Kellomäki1 and Heli 3 Peltola1 4 5 1 School of Forest Sciences, University of Eastern Finland, P.O. B"

_Biogeosciences, 2017_

## Referee Comment (RC1) · Anonymous Referee #1 · 16 May 2017

Review of Gong et al. "Modelling the diurnal and seasonal dynamics of soil CO2 exchange in a semiarid ecosystem with high plant-interspace heterogeneity"

Gong et al. present a model development and model validation study focused on simulating soil CO2 efflux in semiarid soils. They have improved on previous models used for these ecosystems by incorporating abiotic processes related to lateral and vertical transport of CO2 in heterogeneous canopies as well as biotic processes related to biocrust CO2 production and photodegradation. They evaluate their new model against two years of site-based data from semiarid shrubland ecosystem in Yanchi, northwestern China.

This is an interesting and relatively new contribution to the modelling literature on this

topic. The introduction is well laid out and clearly explains the context behind the work and the importance of including processes related to plant heterogeneity and biocrust dynamics in the model. From both the introduction and discussion it is clear the authors know the literature well and have a good handle on the gaps in understanding that need to be addressed. This is a comprehensive study with a number of interesting results.

Given this however, I would like to see some of the objectives framed as questions in the introduction, which would then be answered directly in the results/discussion. This would help to highlight the key points in the results section, link the results back to the context and would make the text less focused on a model description, parameterization and sensitivity study, although these aspects are important and described well in this paper. Such changes would serve to improve the structure, readability and scientific value of the paper. The authors could frame the work around questions such as: ć What is the difference in diurnal to seasonal variability in CO2 flux between soils with and without a biocrust? ć Are there significant differences between the CO2 flux from plant covered and interspace soils? ć What are the relative contributions of different processes to total soil CO2 efflux? Are the process of CO2 production and emission tightly coupled during wetting and drying cycles?

The paper would also benefit from a more thorough discussion of the importance of including processes related to biocrusts in regional to global scale biogeochemical models. Does the inclusion of biocrust-related processes improve the fit to the measured soil CO2 flux at C3 compared to a model that does not include these processes (e.g. if you repeated the simulation without the inclusion of the new processes related to biocrust)? Does this represent a significant flux in semiarid ecosystem C balance at regional to global scales?

Finally, please see my comment below on the aim of the sensitivity study, other than to see how robust the model is to changes in parameters. However, in a more general context, I think it would be beneficial for the study if you put the sensitivity analyses in the context of climate and anthropogenic change? What are the likely changes in

**BGD**
temperature and moisture for this region? And what is the implication for the carbon balance and viability of the vegetation of this ecosystem in the future?

MAIN SUGGESTIONS

Materials and methods

Figure 1: could you incorporate a small map showing the study region? I am sure readers would find that useful. Figure 1 also is not very clear unless I zoom in, therefore I think the size/resolution needs to be improved for reading on paper.

Line 127: You say 40% here but the value is 90% in the Gong et al. (2016) paper. Which one is correct?

Line 160: Please define PATCIS. Is it a model name, an acronym?

Section 2.2.3: for the sake of clarity/completeness, it would be good to have an extra equation here showing how all the flux components sum to provide the total net biocrust flux (FB) in equation (1).

I would have Section 2.3.1 as a separate Section (e.g. just 2.3) entitled "data" or "measurements". It may not be immediately obvious that you would find a description of the data here in this section on model parameterization if you were just scanning through section headings.

Lines 308 to 309: it would be great to have pictures of these three sites to show readers new to the topic of biocrusts what they look like.

Section 2.3.3 (and throughout Section 2.3): For many parameters, there is a clear and adequate description for the functions used to derive them, but not all – some detail appears to be missing for some. For example, for lines 345-347: please could you give a little more detail on how the horizontal and vertical root biomass profiles were parameterized? For example, did the root biomass decrease linearly with distance from the center of the shrub crown? Another example for lines 351-352: how was
the photodegradation coefficient calculated from the mass-loss rate. I am also a bit confused as to why only certain parameters are included in Table 2 and not all (e.g. why is the photodegradation coefficient not included for example)?

Lines 377 and 379: I think the 2nd Q10() in equation 28 should be  $Q10(\theta)$  and the same in equation 30?

How did you come to define equation 31 in this way? Based on the aforementioned studies? Which method did you use to perform the fits shown in Figure 2 and equations 32 and 33?

Section 2.4.1 title should mention the meteorological forcing data. A shorter title could be "model set-up". Do you have a reference for the PECE method?

Lines 502-504: How/why did you choose which parameters to include in your sensitivity analysis?

Section 2.4.3: to avoid some confusion in the results later, I might have the parameter sensitivity as a separate test from the comparison between plant covered and interspace soil CO2 flux (so add a test 5). I think this would help to emphasize the importance of the impact of plant heterogeneity in the text, given its importance in the paper title.

It might be useful for the reader to have a small table summarizing all the tests, which processes they include, which site they correspond to, what the observations are measuring etc.

Results

What is shown on the bottom of Fig 4a? Is that precipitation? It might be worth smoothing your hourly curves with a moving average window so we can see the variability better I would put precipitation on the hourly time series plots in Figure 5 as well.

Lines 540-541: It would be good to give the RMSE of C1 above as well for a compari-
son.

Figure 6: I like the addition of the diurnal bias plots - they are very informative. I would put the same scale for all C1, C2 and C3 plots to enable an easier comparison between the tests.

Lines 542-544: Looking at Figure 6b it seems to me that the pattern of diurnal biases has changed for C3 compared to C2 and C1? There is now a positive bias around noon and a negative bias in the mid-morning and afternoon? Why do you think this is?

Table 3 caption: component fluxes. It would also be better to say "for areas with plant cover and without (interspace)"

Figure 7: You mention FS in the caption but FT in the text and figure legends. Also, you refer to net CO2 sequestration by the biocrust in the legend – isn't this FB (or FCt), of have I misunderstood? It would be helpful to the reader to make sure all the abbreviations you use for the fluxes are uniformly used across the text and figures. In fact, I would suggest adding an extra table with all the component flux abbreviations and their long name/description, given that there are many. This may help the reader cross-reference between the figures, tables and text.

Table 4: are all the values listed the % change in C flux after manipulation compared to the base flux, or are some of the +/- values a change in the absolute C flux magnitude? Please detail this in the table caption.

Lines 578-583: I am a bit lost as to main message of the second part of the sensitivity analysis. What does the sensitivity analysis suggest about how important the parameters are? If changing the parameter values does not result in that much change to the fluxes, does that mean that parameter or even that process is not actually important for modelling the flux? How have you decided how much to change the parameter values? Perhaps it would be good to explore their full range (between their upper and lower bounds) in a proper sensitivity analysis (e.g. using the Morris method) in order
to determine the full impact of the parameter values.

**Discussion**

Some of the results are repeated in the discussion. Given that the results section is very short, it might be better to merge the results and at least some of the discussion that is very pertinent to each particular result for each separate sub-section (e.g. validity of the results, ...) and separate out each with a sub-heading. That way the reader is not switching between different aspects of the modeling in the results before having to come back to consider the implications of these results in the discussion.

It is encouraging that the authors are aware and detail all the caveats of their work; however, the manuscript might benefit from a shorter, more concise discussion, particularly given the methods section is also (necessarily) long. A brief summary of the missing features of the model such as is given in the conclusions may be enough with a few extra sentences and references.

Be sure not to repeat sections of the introduction or results in the discussion, e.g. lines 726-744 is largely a repetition of context and results. I appreciate it is hard to keep the results and discussion separate, which is why I have suggested combining at least some parts of the discussion with the results in a "results and discussion" section. This would also help to reduce the length of the manuscript. Other more general parts of the discussion could be put in a final "Conclusions and future perspectives" section.

**MINOR COMMENTS**

In general: CO2 production, not productions. Both some missing and unnecessary "the" in places (e.g. line 95, no 'The' is needed, and occasionally the plural of a word is used where it should not be (e.g. CO2 productions). Check the text carefully.

Please could you explain this sentence more: "In dryland soils, the interactions between CO2 transport and water cycle could also be intensive, due to the commonly high salinity/alkalinity of soils."? What do you mean by intensive?

**BGD**
Line 104: "However, both models focus on the patterns at the regional-scale with very simplified ecosystem processes and neglect stand-scale heterogeneities of waterenergy budget, and have not yet been validated by field measurements." I would turn this sentence into a positive one to highlight what you will do to add to the field and incorporate that into your following paragraph. Something like switching this sentence to read "we will build on this work by including complex processes related to.... Furthermore, we have validated our new model with extensive field measurements..."

Line 384: Sponseller, 2007 and Cable et al. references missing

Lines 409-411: Do you mean NPP and not NP?

Line 429: litterfall

Line 456: probability, not probably?

Line 464: "The model simulation employed half-hourly meteorological factors" "the model was run with half-hourly meteorological variables"

Line 496: "contributed to the soil CO2..."?

Line 506: "It was also studied" "Furthermore, we studied"

Line 507: "regarding the" "due to"

Line 521: pronounced

Line 703: "should" instead of "shall"

Lines 705-709 reads more like "Conclusions"

**BGD**

---

## Referee Comment (RC2) · R. Scott (Referee) · 2 Jun 2017

This study concerns the development and application of a highly detailed physically based patch-scale land-atmosphere energy, water and carbon balance model for a semiarid ecosystem with high plant-interspace heterogeneity. The model represents an expansion of the the model developed by Gong et al. (2016. Ag. Forest Met) that compared patch scale water and energy exchange into soil-plant C exchanges. The model represents most of the C stocks and fluxes that you expect to be relevant for dryland ecosystems, but which are not normally represented in ecosystem C models like photodegradation, biocrust photosynthesis and respiration, gas and liquid phase

co2 transport, etc. The model is used to simulate bare and plant shaded surfaces as well as biocrust covered surfaces and compared to measurements.

The model was shown to be very capable of accurately simulating measured soil temperatures, soil moisture and soil respiration (Rs). The main findings were that total CO2 production in the soil and Rs could deviate substantially from one another due to root uptake, crust respiration and photosynthesis, and variations in CO2 dissolution, emphasizing the processes beyond heterotrophic and autotrophic respiration and highly heterogeneous nature of CO2 cycling in patchy ecosystems. These results shed some light on the importance of these other processes that are not commonly represented in ecosystem models as well as our ability to represent them in ecosystem models.

The paper is well written. The authors do a great job in discussing the background literature in the Introduction as well as tying their findings to previous studies in the discussion. The paper is very long, but this should probably be expected given the highly detailed modeling work that is being presented. Overall, there is nothing fundamentally flawed with the paper and I expect that this work will be of interest to ecosystem modelers, particularly those interested in dryland ecosystems.

My main complaint about the paper involves equifinality of the model results and the lack of data to be able to validate their findings on the relative roles of the different component fluxes. A model with far fewer parameters and processes is likely to be equally as capable of simulating soil moisture, temperature, and Rs for these cases or tests so how can one have much confidence that extra capabilities of the model (to represent the individual fluxes and transports like photorespiration, crust photosynthesis/respiration, CO2 uptake by roots) are valid? Table 3 is great, but it could be entirely fictitious. While I'm excited to see models being built with these processes considered, I'm wonder how we can build confidence that they are any better than simple, more empirical models already out there.

A couple of relatively minor issues: 1. There could be better setup in the introduction.

What are the objectives and rationale of this study? Major questions or hypotheses? 2. Figure 8 used as an estimate of photorespiration. How do you separate the effects of greenhouse effect under the clear chamber versus the shade effect of the opaque one? In other words, the opaque chamber shields the surface and reduces the heating when the chamber is closed. The clear chamber, by allowing solar radiation in and blocking thermal radiation out, is going to be heated much more potentially during the measurement cycle, potentially increasing heterotrophic respiration. Is Rs higher because of higher temps or because of photorespiration?

Text specific comments: L13. This sentence is an unusual way to open up an Abstract. I am wondering if it could be replaced with a sentence that provides context and rationale for the study L54. cannot L55. periods L58 intensive? Also, why would water and CO2 transport be more intensive in the drylands? L62. Here's another paper with the relevance of abiotic C with fluxes on the diurnal time scale. Hamerlynck, Erik P., et al. "Nocturnal soil CO2 uptake and its relationship to subsurface soil and ecosystem carbon fluxes in a Chihuahuan Desert shrubland." Journal of Geophysical Research: Biogeosciences 118.4 (2013): 1593-1603.. There are several papers out that seem to show that inorganic C uptake is unlikely to be a very significant flux...see e.g. review in Schlesinger, William H. "An evaluation of abiotic carbon sinks in deserts." Global change biology 23.1 (2017): 25-27. L69. matter L70. "could maintain inactive"? L77. Might consider H. Throop's work here, e.g., Throop, Heather L., and Steven R. Archer. "Resolving the dryland decomposition conundrum: some new perspectives on potential drivers." Progress in botany. Springer Berlin Heidelberg, 2009. 171-194. If you can't find this chapter, she has several articles about photodegradation. L79. periods L95 Define "global change" L104. "works" L114-115. This sentence seems out of place. If this represents an advance of Gong et al. you should cover what this model development is. L118. How about "Model Overview" L119. modeling work was based on measurements? L128. Don't understand this sentence L456 "probably"? L491. Later on, Test 4 is mentioned, but it should probably be included in this paragraph L503. component L521. "was more pronounced" L523-525. I don't know of many soil

water probes that are good at measuring frozen water content. Are you sure the measurements are valid during these times? L534 4b? L554. All the variables need to be clearly redefined in the Table caption so that this paragraph is much easier to understand. L565. Compared to L566 ,irrespective of the size... L573. "compared" L598. "our model capably reproduced the time series for the water and energy fluxes at ..." L605 Suggest using another heading before this paragraph, something like "modeling uncertainties" L705. provides L708 caution L709 Our simulations showed that a L773. uptake L785 Are the model and data available for others to use ? Table 3. All terms need to be defined in this table caption including Fs, Fft PCt Figure. 1. The photo is really too small to see much of anything. Suggest deleting this so there is more space for the conceptual figure Fig. 4. Ppt is not labeled or given a scale Fig. 6. Greek letter is not defined in the caption. Fig. 7. This figure is very hard to see. Could you use more colors for the different symbols so that it is easier to see?

---

## Author Comment (AC1) · 15 Jul 2017

General comment "Modelling the diurnal and seasonal dynamics of soil CO2 exchange in a semiarid ecosystem with high plant-interspace heterogeneity" Gong et al. present a model development and model validation study focused on simulating soil CO2 efīňĆux in semiarid soils. They have improved on previous models used for these ecosystems by incorporating abiotic processes related to lateral and vertical transport of CO2 in heterogeneous canopies as well as biotic processes related to biocrust CO2 production and photodegradation. They evaluate their new model against two years of site-based data from semiarid shrubland ecosystem in Yanchi, northwestern China.

This is an interesting and relatively new contribution to the modelling literature on this topic. The introduction is well laid out and clearly explains the context behind the work and the importance of including processes related to plant heterogeneity and biocrust dynamics in the model. From both the introduction and discussion it is clear the authors know the literature well and have a good handle on the gaps in understanding that need to be addressed. This is a comprehensive study with a number of interesting results. Given this however, I would like to see some of the objectives framed as questions in the introduction, which would then be answered directly in the results/discussion. This would help to highlight the key points in the results section, link the results back to the context and would make the text less focused on a model description, parameterization and sensitivity study, although these aspects are important and described well in this paper. Such changes would serve to improve the structure, readability and scientiiňAc value of the paper. The authors could frame the work around questions such as: (a) What is the difference in diurnal to seasonal variability in CO2 indux between soils with and without a biocrust? (b) Are there signijnAcant differences between the CO2 inCux from plant covered and interspace soils? (c) What are the relative contributions of different processes to total soil CO2 efincture? Are the process of CO2 production and emission tightly coupled during wetting and drying cycles? The paper would also beneïňAt from a more thorough discussion of the importance of including processes related to biocrusts in regional to global scale biogeochemical models. Does the inclusion of biocrust-related processes improve the *iňAt* to the measured soil CO2 *iňCux* at C3 compared to a model that does not include these processes (e.g. if you repeated the simulation without the inclusion of the new processes related to biocrust)? Does this represent a signiiňAcant iňĆux in semiarid ecosystem C balance at regional to global scales? Finally, please see my comment below on the aim of the sensitivity study, other than to see how robust the model is to changes in parameters. However, in a more general context, I think it would be beneïňAcial for the study if you put the sensitivity analyses in the context of climate and anthropogenic change? What are the likely changes in temperature and moisture for this region? And what is the implication

**BGD**
for the carbon balance and viability of the vegetation of this ecosystem in the future? "

Response to general comment: We are very grateful for the efforts of reviewer on improving this work. Indeed, this paper has been organized as modelling-oriented. The main motivation is that, although there has been many studies and data on different flux components (they all seems very important somehow), there hasn't been any system that could integrated those knowledge, or a "playground" to explain the C dynamics or make extrapolations, e.g. to a different space-time or scenario. Therefore, model development became the primary objective. We do agree that, for results and discussion, better structuring is very much in need. Within the reach of this work, we have re-organized the analysis around two aspects, i.e. i) the roles of componential C processes in regulating soil CO2 effluxes, and ii) the plant-interspace differences in the C fluxes. Sections and paragraphs in results and discussion has been rearranged accordingly, and extra comparison (e.g. Table 5) has been added to aid the second guestion. We also performed a test as suggested, to see if exclusion of biocrust-related processes reduced the iňAt to the measured soil CO2 iňĆux at C3 (Line 676). We hope these revisions could help the reader to better understand the emphasis and the findings. Please find the red marks in the supplementary file for those revised places.

We understand the excitement of reviewer to upscale and extrapolate the current results, for large scale, long term and general implications. Extrapolating the modelling for large scale and long term applications, and scenarios analysis for climate change and sustainable management are our future purposes as well. However, these discussions are largely out of the reach of current model. For example, as described in section 4.3, the growth dynamics of shrub vegetation is not included in modelling yet. Therefore, the changes in leaf area, shading and energy partitioning, evapotranspiration and root biomass are largely unknown and hard to set for longer term iterations, climate change scenarios or vegetation management. Our ongoing work addresses this issue and trying to complete the full picture of C-N cycle in such ecosystems. Then we might have a better stand for deeper discussions on those topics. BGD
MAIN SUGGESTIONS Materials and methods Figure 1: could you incorporate a small map showing the study region? I am sure readers would in And that useful. Figure 1 also is not very clear unless I zoom in, therefore I think the size/resolution needs to be improved for reading on paper.

Response: We appreciate the comment and has separated Figure 1 to two different figures (Fig. 1 and 2). In Fig. 1, we added a map showing the study region (Fig. 1a), changed the site picture (Fig. 1b) for better quality, and added photos showing the soil collars (Fig. 1c - 1e). Model framework has been moved to Fig. 2.

Line 127: You say 40% here but the value is 90% in the Gong et al. (2016) paper. Which one is correct?

Response: Both. This study was based on a different location from that in Gong et al. (2006). The two locations are about 1km apart.

Line 160: Please deïňĄne PATCIS. Is it a model name, an acronym?

Response: It is the name of model.

Section 2.2.3: for the sake of clarity/completeness, it would be good to have an extra equation here showing how all the īňĆux components sum to provide the total net biocrust īňĆux (FB) in equation (1).

Response: We agree that the flux symbolism is somehow quite messy. Therefore, we have revised all the symbols and names to keep them consistent. FB actually should be FT in this case.

I would have Section 2.3.1 as a separate Section (e.g. just 2.3) entitled "data" or "measurements". It may not be immediately obvious that you would iňĄnd a description of the data here in this section on model parameterization if you were just scanning through section headings.

Response: Good suggestion. We have separated that paragraph to section 2.3 named

**BGD**
" Micrometeorological and soil CO2 efflux measurement".

Lines 308 to 309: it would be great to have pictures of these three sites to show readers new to the topic of biocrusts what they look like.

Response: The crust pictures have added to Fig. 1 (c - e).

Section 2.3.3 (and throughout Section 2.3): For many parameters, there is a clear and adequate description for the functions used to derive them, but not all – some detail appears to be missing for some. For example, for lines 345-347: please could you give a little more detail on how the horizontal and vertical root biomass proiňAles were parameterized? For example, did the root biomass decrease linearly with distance from the center of the shrub crown? Another example for lines 351-352: how was the photodegradation coefiňAcient calculated from the mass-loss rate. I am also a bit confused as to why only certain parameters are included in Table 2 and not all (e.g. why is the photodegradation coefiňAcient not included for example)?

Response: That section (numbered as 2.4.2 in revised paper) has been checked and more information has been added. Photodegradation coefficient kp was indeed missing from Table 2 and now has been added.

Lines 377 and 379: I think the 2nd Q10() in equation 28 should be Q10( $\theta$ ) and the same in equation 30? How did you come to deïňĄne equation 31 in this way? Based on the aforementioned studies? Which method did you use to perform the ïňĄts shown in Figure 2 and equations 32 and 33?

Response: The 2nd Q10() in equation 28 has been reivsed to Q10( $\theta$ ). For equation 31, actually there were no available numerical descriptions on such an rain effect, therefore we decided to add one to the algorithm. This equation has been tested in sensitivity analysis (see test for parameter np), which shows this equation may not be an important source of uncertainty. Future modelling may also consider to simplify this algorithm (Line 609). Fittings in figure 2 (Fig. 3 in revised manuscript) were performed
by Matlab curve-fitting toolbox. The information has been added to Line 410 and 417.

Section 2.4.1 title should mention the meteorological forcing data. A shorter title could be "model set-up". Do you have a reference for the PECE method?

Response: The title has been changed as suggested (Line 322). Like forward/backward Euler, PECE method can be found in many textbooks related to ordinary differential equations, e.g. Butcher, John C. (2003), Numerical Methods for Ordinary Differential Equations, New York: John Wiley & Sons, ISBN 978-0-471-96758-3.

Lines502-504: How/why did you choose which parameters to include in your sensitivity analysis?

Response: The reason to choose the tested parameters has been better demonstrated in section 2.5.3.

Section 2.4.3: to avoid some confusion in the results later, I might have the parameter sensitivity as a separate test from the comparison between plant covered and interspace soil CO2 inĆux (so add a test 5). I think this would help to emphasize the importance of the impact of plant heterogeneity in the text, given its importance in the paper title. It might be useful for the reader to have a small table summarizing all the tests, which processes they include, which site they correspond to, what the observations are measuring etc.

Response: This suggestion has been taken with gratitude. We separated the contents as suggested, and organized the tests to two part: i) to demonstrate the roles of componential C fluxes in regulating surface efflux; and ii) to find how the plant cover and interspace are different in the flux rates and sensitivities. see Section 2.5.3-2.5.4.

Results What is shown on the bottom of Fig4a? Is that precipitation? It might be worth smoothing your hourly curves with a moving average window so we can see the variability better I would put precipitation on the hourly time series plots in Figure 5 as well.
Response: The figure 4 and 5 has been revised following the suggestions. 3-day moving average trends were used to show the temporal dynamics better.

Lines 540-541: It would be good to give the RMSE of C1 above as well for a comparison.

Response: The figure 4 and 5 has been revised as suggested. 3-day moving average trends were used to show the temporal dynamics better.

Figure 6: I like the addition of the diurnal bias plots - they are very informative. I would put the same scale for all C1, C2 and C3 plots to enable an easier comparison between the tests.

Response: The scales has been set to same in Fig. 5 and Fig.7.

Lines 542-544: Looking at Figure 6b it seems to me that the pattern of diurnal biases has changed for C3 compared to C2 and C1? There is now a positive bias around noon and a negative bias in the mid-morning and afternoon? Why do you think this is?

Response: Indeed, the pattern became different in C3 compared to C2 and C1. It is probably caused by biases from photosynthesis & photodegradation, which were introduced to system in Test 3. We have corrected the description in the manuscript (Line 560-563).

Table 3 caption: component ïňĆuxes. It would also be better to say "for areas with plant cover and without (interspace)"

Response: Table 3 caption has been revised as suggested.

Figure 7: You mention FS in the caption but FT in the text and ïňAgure legends. Also, you refer to net CO2 sequestration by the biocrust in the legend – isn't this FB (or FCt), of have I misunderstood? It would be helpful to the reader to make sure all the abbreviations you use for the ïňĆuxes are uniformly used across the text and ïňAgures. In fact, I would suggest adding an extra table with all the component ïňĆux abbreviations
and their long name/description, given that there are many. This may help the reader cross-reference between the ïňAgures, tables and text.

Response: There have been several place with unnecessary naming which has complicated the whole thing. We have revised all the symbols and names to keep them consistent. Please see in the revised figure (Fig. 8).

Table 4: are all the values listed the % change in C ĭňĆux after manipulation compared to the base ĩňĆux, or are some of the +/- values a change in the absolute C ĩňĆux magnitude? Please detail this in the table caption.

Response: Yes all values listed in Table 4 are % changes. This point has been clarified in figure footnote.

Lines 578-583: I am a bit lost as to main message of the second part of the sensitivity analysis. What does the sensitivity analysis suggest about how important the parameters are? If changing the parameter values does not result in that much change to the ïňĆuxes, does that mean that parameter or even that process is not actually important for modelling the ïňĆux? How have you decided how much to change the parameter values? Perhaps it would be good to explore their full range (between their upper and lower bounds) in a proper sensitivity analysis (e.g. using the Morris method) in order to determine the full impact of the parameter values.

Response: Yes the analysis of parameter sensitivity is to understand which parameter is more important and more likely to be the main source of uncertainty, as we have many site-specific parameters. Those parameter of high sensitivity then need to be use with extra cautious, when applying the system to another space-time. For sitespecific parameters like Ts,  $\theta$ , Mtot etc., we modified the values by the same degree (±10%), so that their effects on C fluxes are easy to compare. This is a bit different in purpose than the Morris method. On the other hand, "full" impact is difficult to define as well. Some parameters, e.g. root biomass, may vary by several folds from one collar to another (see Wang et al., 2015, Biogeosciences). Also an artificial parameter

**BGD**
setting (e.g. assuming extreme values for many parameters) may seem unreal (e.g. a combination of very low moisture content and very high root biomass). Therefore, performing an Morris analysis like suggested could be difficult, while the upper/lower bounds of parameters are unclear in a combination.

Some of the results are repeated in the discussion. Given that the results section is very short, it might be better to merge the results and at least some of the discussion that is very pertinent to each particular result for each separate sub-section (e.g. validity of the results, ...) and separate out each with a sub-heading. That way the reader is not switching between different aspects of the modeling in the results before having to come back to consider the implications of these results in the discussion. It is encouraging that the authors are aware and detail all the caveats of their work; however, the manuscript might beneïňAt from a shorter, more concise discussion, particularly given the methods section is also (necessarily) long. A brief summary of the missing features of the model such as is given in the conclusions may be enough with a few extra sentences and references. Be sure not to repeat sections of the introduction or results in the discussion, e.g. lines 726-744 is largely a repetition of context and results. I appreciate it is hard to keep the results and discussion separate, which is why I have suggested combining at least some parts of the discussion with the results in a "results and discussion" section. This would also help to reduce the length of the manuscript. Other more general parts of the discussion could be put in a in Anal "Conclusions and future perspectives" section.

Response: We greatly appreciate these advices. In order to better structure this part, we combined apart of the discussion on model validity with the result section 3.1, and let the rest discussion part (i.e. section 4.1 and 4.2) to focus on answering the two questions we proposed. The model uncertainties are discussed in the final section 4.3. Still, we would like to provide a deeper and more thorough discussion on model uncertainties and challenges instead of a general and brief one, in order to be more precise and informative about the problems we haven't solved, or those could be important to

**BGD**
further studies.

MINOR COMMENTS In general: CO2 production, not productions. Both some missing and unnecessary "the" in places (e.g. line 95, no 'The' is needed, and occasionally the plural of a word is used where it should not be (e.g. CO2 productions). Check the text carefully. Please could you explain this sentence more: "In dryland soils, the interactions between CO2 transport and water cycle could also be intensive, due to the commonly high salinity/alkalinity of soils."? What do you mean by intensive?

Response: Appreciated. We checked all possibly mistaken forms in revised manuscript. The "dryland soil" sentence has been revised, see Line 59-61.

Line 104: "However, both models focus on the patterns at the regional-scale with very simpliñĂAed ecosystem processes and neglect stand-scale heterogeneities of water energy budget, and have not yet been validated by ĩňAeld measurements." I would turn this sentence into a positive one to highlight what you will do to add to the ĩňAeld and incorporate that into your following paragraph. Something like switching this sentence to read "we will build on this work by including complex processes related to.... Furthermore, we have validated our new model with extensive ĩňAeld measurements..."

Response: we have revised the part as suggested, see Line 107-109.

Line 384: Sponseller, 2007 and Cable et al. references missing

Response: Citation to Sponseller has been added. Cable et al. 2013 was removed.

Lines 409-411: Do you mean NPP and not NP?

Response: Yes, it should be NPP here. Corrected.

Line 429: litterfall Line 456: probability, not probably?

Response: Corrected.

Line 464: "The model simulation employed half-hourly meteorological factors" "the
model was run with half-hourly meteorological variables"

Response: Corrected as suggested (Line 468).

Line 496: "contributed to the soil CO2..."?

Response: Corrected (Line 499).

Line 506: "It was also studied" "Furthermore, we studied" Line 507: "regarding the" "due to"

Response: This paragraph has been revised (section 2.5.3).

Line 521: pronounced

Response: Revised as suggested (Line 536).

Line 703: "should" instead of "shall"

Response: This paragraph has been revised.

Lines 705-709 reads more like "Conclusions"

Response: This section has been revised.

Please also note the supplement to this comment: https://www.biogeosciences-discuss.net/bg-2017-95/bg-2017-95-AC1-supplement.zip

---

## Author Comment (AC2) · 15 Jul 2017

General comment: "This study concerns the development and application of a highly detailed physically based patch-scale land-atmosphere energy, water and carbon balance model for a semiarid ecosystem with high plant-interspace heterogeneity. The model represents an expansion of the model developed by Gong et al. (2016. Ag. Forest Met) that compared patch scale water and energy exchange into soil-plant C exchanges. The model represents most of the C stocks and fluxes that you expect to be relevant for dryland ecosystems, but which are not normally represented in ecosystem C models like photodegradation, biocrust photosynthesis and respiration, gas and liquid phase co2 transport, etc. The model is used to simulate bare and plant shaded surfaces as well as biocrust covered surfaces and compared to measurements. The model was shown to be very capable of accurately simulating measured soil temperatures, soil moisture and soil respiration(Rs). The main findings were that totalCO2 production in the soil and Rs could deviate substantially from one another due to root uptake, crust respiration and photosynthesis, and variations in CO2 dissolution, emphasizing the processes beyond heterotrophic and autotrophic respiration and highly heterogeneous nature of CO2 cycling in patchy ecosystems. These results shed some light on the importance of these other processes that are not commonly represented in ecosystem models as well as our ability to represent them in ecosystem models. The paper is well written. The authors do a great job in discussing the background literature in the Introduction as well as tying their findings to previous studies in the discussion. The paper is very long, but this should probably be expected given the highly detailed modeling work that is being presented. Overall, there is nothing fundamentally flawed with the paper and I expect that this work will be of interest to ecosystem modelers, particularly those interested in dryland ecosystems. My main complaint about the paper involves equifinality of the model results and the lack of data to be able to validate their findings on the relative roles of the different component fluxes. A model with far fewer parameters and processes is likely to be equally as capable of simulating soil moisture, temperature, and Rs for these cases or tests so how can one have much confidence that extra capabilities of the model (to represent the individual fluxes and transports like photorespiration, crust photosynthesis/respiration, CO2 uptake by roots) are valid? Table 3 is great, but it could be entirely fictitious. While I'm excited to see models being built with these processes considered, I'm wonder how we can build confidence that they are any better than simple, more empirical models already out there. "

Response to general comment:

We sincerely appreciate the hard work of reviewer and sharp comments. The

manuscript has been revised in light of the comments, with several main changes in the structure, figures and tables. All the modified texts have been marked in red color in the revised manuscript. See the changes in text, figures and tables in supplementary file.

For the main complaint of the reviewer, we fully aware that over-complexity and over-parameterization could be important sources of uncertainties for process-based models. However, good fitting may not be the ultimate goal of modelling. Mechanistic models are found on mimicking the system structure and processes, breaking the big black box into smaller and simpler ones (which is also easier to experiment on) and connecting them by known cause-effects, so that it could integrate existing knowledge and possibly make some extrapolation to a different space-time. I must emphasize that a well-trained regressive model with much less parameters may have high goodness-of-fitting, but does not necessarily explain how an ecosystem works, or clarify the scope of its applicability – so we might argue if it is the suitable way to apply Occam's razor. For example, one may need enormous empirical models to calculate CO2 emissions, in order to cover different combination of environmental factors, soil properties, canopy features and biocrust types. However, through incorporating different modules (processes) and parameter values, mechanistic modelling actually serves a simpler and more rational way to aid this complication. Moreover, mechanistic models (like this one) are eventually found on small "black boxes", at which level detailing the mechanisms further become difficult and using simple empirical functions become near-optimal. In this sense, Occam's razor still applies, and stays with the concept of the mechanistic model.

We also keep in mind about the uncertainties of modelling. To separate the individual fluxes, we used multiple chambers (C1, without crust influences; C2, with only dark respiration; C3 with photosynthesis and photodegradation) to perform a step-wise validation. However, as pointed out by reviewer, only one chamber for each step may not be enough and fluxes like photorespiration and CO2 uptake by root are still lack of support from data. We are planning more measurements addressing these issues. These uncertainties, along with several other possible aspects, have been demonstrated in revised section 4.3 "Uncertainties and challenges".

"A couple of relatively minor issues: 1. There could be better setup in the introduction. What are the objectives and rationale of this study? Major questions or hypotheses? "

Response to minor issue 1: This setup of introduction may not be optimal for generating questions and hypotheses. However, we decided to bring up the work from the view of modelling, as the main problem is that there hasn't been any method, so far, that for researchers could integrate those most discussed C processes for dryland ecosystem. Without such a "playground" in the first place, generating questions and hypothesis regarding the componential fluxes and subscale heterogeneities, or their environmental sensitivities, will be difficult. In this case, model development become a primary objective, and this has been emphasized particularly in Line 96-108 of introduction. For better demonstrate the modelling results and tests, we re-organized the result section and tried to investigate two specific questions: i) the roles of componential C processes in regulating soil $CO_2$ effluxes in the studied ecosystem, and ii) the plant-interspace differences in the componential C processes. These contents has been added to introduction as well, see Line 115-117.

"2. Figure 8 used as an estimate of photorespiration. How do you separate the effects of greenhouse effect under the clear chamber versus the shade effect of the opaque one? In other words, the opaque chamber shields the surface and reduces the heating when the chamber is closed. The clear chamber, by allowing solar radiation in and blocking thermal radiation out, is going to be heated much more potentially during the measurement cycle, potentially increasing heterotrophic respiration. Is Rs higher because of higher temps or because of photorespiration? "

Response to minor issue 2: The C fluxes are measured by automatic chambers, which only seals the collar during measurement (2.5 minutes), then move away from the

collar. Therefore, the collar was not blocked by chamber in most of time, and the temperature disturbance by measurement are marginal. As suggested by Figure 8, flux signals during the daytime nearly doubled during those periods; it is difficult to be explained as heating effect, as even 2 degree heating lead to < 10% changes in efflux (Table 4, sensitivity analysis). Also, the period was dry and with almost no rain event (see Fig. 6 in revised manuscript). Therefore, photorespiration by crust organisms is also unlikely.

"Text specific comments: L13. This sentence is an unusual way to open up an Abstract. I am wondering if it could be replaced with a sentence that provides context and rationale for the study "

Response: The abstract opening has been revised (Line 13-16).

L54. cannot

Response: Revised to "may not" (Line 56)

L55. periods

Response: Revised to "periods" (Line 58)

L58 intensive? Also, why would water and CO2 transport be more intensive in the drylands?

Response: We are agree with reviewer that this claim could be assertive. The sentence has been revised to "In dryland soils of high salinity/alkalinity, CO2 transport and water cycle are tightly coupled, as large inorganic C fluxes can be driven solely by dissolution and infiltration of CO2 and carbonates" (Line 59).

L62. Here's another paper with the relevance of abiotic C with ïnĆuxes on the diurnal time scale. Hamerlynck, Erik P., et al. "Nocturnal soil CO2 uptake and its relationship to subsurface soil and ecosystem carbon ïnĆuxes in a Chihuahuan Desert shrubland." Journal of Geophysical Research: Biogeosciences 118.4 (2013): 1593-1603.. There

are several papers out that seem to show that inorganic C uptake is unlikely to be a very significant flux...see e.g. review in Schlesinger, William H. "An evaluation of abiotic carbon sinks in deserts." Global change biology 23.1 (2017): 25-27.

Response: It is true that inorganic C uptake may not be very significant flux in many cases. Our simulation also showed that such a flux was only about 15% of total emission from collar, and those C may still emitted somewhere during the transport. The main idea to include the transport processes are to better explain the variations of efflux, which may not necessarily caused by changes in soil C pool, but just caused by noises from the transportation process.

L69. matter Response: Revised as suggested.

L70. "could maintain inactive"? Response: Revised to "could be inactive".

L77. Might consider H. Throop's work here, e.g., Throop, Heather L., and Steven R. Archer. "Resolving the dryland decomposition conundrum: some new perspectives on potential drivers." Progress in botany. Springer Berlin Heidelberg, 2009. 171-194. If you can't find this chapter, she has several articles about photodegradation.

Response: Good suggestion. Citation of Throop et al. 2009 has been added.

L79. periods Response: Revised as suggested (Line 80)

L95 Define "global change" Response: Revised to "global climate change" (Line 96)

L104. "works" Response: Revised as suggested (Line 105)

L114-115. This sentence seems out of place. If this represents an advance of Gong et al. you should cover what this model development is. Response: This paragraph has been revised.

L118. How about "Model Overview" Response: Revised as suggested (Line 118)

L119. modeling work was based on measurements? Response: Reworded as "...

model was build based on ... " (Line 119)

L128. Don't understand this sentence Response: The sentence has been reworded (Line 128).

L456 "probably"? Response: Revised as suggested (Line 460).

L491. Later on, Test 4 is mentioned, but it should probably be included in this paragraph Response: Test 1-3 are for model validation, Test 4-5 are for sensitivity analysis, so it could be better to separate them into different sections.

L503. component Response: This paragraph has been revised.

L521. "was more pronounced" Response: Revised as suggested (Line 536).

L523-525. I don't know of many soil C3 water probes that are good at measuring frozen water content. Are you sure the measurements are valid during these times? Response: It is true that water content measurement during freezing period may not be reliable. We have changed the statement here (Line 537).

L534 4b? Response: revised to 5b (Line 549).

L554. All the variables need to be clearly redefiˌned in the Table caption so that this paragraph is much easier to understand. Response: Table 3 has been revised and the definition of variables has been added.

L565. Compared to Response: The paragraph has been revised.

L566 ,irrespective of the size... Response: Revised as suggested (Line 583).

L573. "compared" Response: Revised as suggested (Line 599).

L598. "our model capably reproduced the time series for the water and energy fluxes at ..." Response: Revised as suggested (Line 542).

L605 Suggest using another heading before this paragraph, something like "modeling uncertainties" Response: This has been suggested by both reviewer and we have

reorganized this part to section 4.3

L705. provides L708 caution L709 Our simulations showed that a Response: The result section has been reorganized. Those sentences has been rewritten or removed.

L773. uptake Response: revised as suggested (Line 783).

L785 Are the model and data available for others to use ? Response: So far, it is among several collaborators but yes. We are still trying to include the aboveground vegetation and develop the system further.

Table 3. All terms need to be defi̧ned in this table caption including Fs, Fft PCt Response: Revised. Definitions have been added to table footnote.

Figure. 1. The photo is really too small to see much of anything. Suggest deleting this so there is more space for the conceptual fi̧gure Response: Revised. The conceptual framework has been displayed separately as Fig. 2.

Fig. 4. Ppt is not labeled or given a scale Response: Revised (see Fig. 5).

Fig. 6. Greek letter is not defi̧ned in the caption. Response: Revised (see Fig. 7).

Fig. 7. This fi̧gure is very hard to see. Could you use more colors for the different symbols so that it is easier to see? Response: We revised the figure by differentiating the coloring (Figure 8). The resolution of initial figure was also limited by the file-size restrictions of discussion paper. We will upload bigger images with better qualities in the final submission.

Please also note the supplement to this comment:
https://www.biogeosciences-discuss.net/bg-2017-95/bg-2017-95-AC2-supplement.zip

---

## Author Response (AR1)

Final Response to reviewer #1

General comment

*"Modelling the diurnal and seasonal dynamics of soil CO2 exchange in a semiarid ecosystem with high plant-interspace heterogeneity" Gong et al. present a model development and model validation study focused on simulating soil CO2 efflux in semiarid soils. They have improved on previous models used for these ecosystems by incorporating abiotic processes related to lateral and vertical transport of CO2 in heterogeneous canopies as well as biotic processes related to biocrustCO2productionandphotodegradation. They evaluate their new model against two years of site-based data from semiarid shrubland ecosystem in Yanchi, northwestern China. This is an interesting and relatively new contribution to the modelling literature on this topic. The introduction is well laid out and clearly explains the context behind the work and the importance of including processes related to plant heterogeneity and biocrust dynamics in the model. From both the introduction and discussion it is clear the authors know the literature well and have a good handle on the gaps in understanding that need to be addressed.*

*This is a comprehensive study with a number of interesting results. Given this however, I would like to see some of the objectives framed as questions in the introduction, which would then be answered directly in the results/discussion. This would help to highlight the key points in the results section, link the results back to the context and would make the text less focused on a model description, parameterization and sensitivity study, although these aspects are important and described well in this paper. Such changes would serve to improve the structure, readability and scientific value of the paper. The authors could frame the work around questions such as: (a) What is the difference in diurnal to seasonal variability in CO2 flux between soils with and without a biocrust? (b) Are there significant differences between the CO2 flux from plant covered and interspace soils? (c) What are the relative contributions of different processes to total soil CO2 efflux? Are the process of CO2 production and emission tightly coupled during wetting and drying cycles? The paper would also benefit from a more thorough discussion of the importance of including processes related to biocrusts in regional to global scale biogeochemical models. Does the inclusion of biocrust-related processes improve the fit to the measured soil CO2 flux at C3 compared to a model that does not include these processes (e.g. if you repeated the simulation without the inclusion of the new processes related to biocrust)? Does this represent a significant flux in semiarid ecosystem C balance at regional to global scales? Finally, please see my comment below on the aim of the sensitivity study, other than to see how robust the model is to changes in parameters. However, in a more general context, I think it would be beneficial for the study if you put the sensitivity analyses in the context of climate and anthropogenic change? What are the likely changes in temperature and moisture for this region? And what is the implication for the carbon balance and viability of the vegetation of this ecosystem in the future?"*

Response to general comment:

We are very grateful for the efforts of reviewer on improving this work. We have re-organized the objectives and contents, emphasizing on the following two questions: i) the roles of componential C processes in regulating soil CO2 effluxes, and ii) the plant-interspace differences in the C fluxes (see Lines 111-120). Sections and paragraphs in results and discussion has been re-arranged accordingly, and additional analysis (e.g. Table 5) has been added to aid the second question. We also performed a test as suggested, to see if exclusion of biocrust-related processes reduced the fitness to measured soil CO2 flux at C3 (Line 678-683). We hope these revisions could improve the manuscript. Please find the red marks in revised manuscript for the changes.

We understand the hope of reviewer to upscale and extrapolate the current results, for large scale, long term and general implications. Extrapolating the modelling for large scale and long term applications, and scenarios analysis for climate change and sustainable management are our future targets as well. At the moment, this could not be done by current model. Because, as described in section 4.3, the growth dynamics of shrub vegetation is not included in modelling yet. Therefore, the changes in leaf area, shading and energy partitioning, evapotranspiration and root biomass are not available for longer term iterations, considering climate change scenarios or vegetation management. Our future work addresses this issue and will aim to provide a full picture of C-N cycle in such ecosystems.

*MAIN SUGGESTIONS Materials and methods Figure 1: could you incorporate a small map showing the study region? I am sure readers would find that useful. Figure1 also is not very clear unlessI zoom in, therefore I think the size/resolution needs to be improved for reading on paper.*

Response: We appreciate the comment and have separated Figure 1 into two different figures (Fig. 1 and 2). Into Fig. 1, we added a map showing the study region (Fig. 1a), changed the site picture (Fig. 1b) for better quality, and added photos showing the soil collars (Fig. 1c - 1e). Model framework has been moved to Fig. 2.

*Line 127: You say 40% here but the value is 90% in the Gong et al. (2016) paper. Which one is correct?*

Response: Both values are corrent. This study was based on a different location from that in Gong et al. (2006). The two locations are about 1km apart.

*Line 160: Please define PATCIS. Is it a model name, an acronym?*

Response: It is the name of model.

*Section 2.2.3: for the sake of clarity/completeness, it would be good to have an extra equation here showing how all the flux components sum to provide the total net biocrust flux (FB) in equation (1).*

Response: We have revised all the symbols and names to keep them consistent. $F_B$ in this question was replaced by $F_{Ct}$, see Line 162.

*I would have Section 2.3.1 as a separate Section (e.g. just 2.3) entitled "data" or "measurements". It may not be immediately obvious that you would find a description of the data here in this section on model parameterization if you were just scanning through section headings.*

Response: We have separated that paragraph to section 2.3 named " Micrometeorological and soil CO2 efflux measurement". (Line 299-324)

*Lines 308 to 309: it would be great to have pictures of these three sites to show readers new to the topic of biocrusts what they look like.*

Response: The crust pictures have added to Fig. 1 (c - e).

*Section 2.3.3 (and throughout Section 2.3): For many parameters, there is a clear and adequate description for the functions used to derive them, but not all – some detail appears to be missing for some. For example,*

*for lines 345-347: please could you give a little more detail on how the horizontal and vertical root biomass profiles were parameterized? For example, did the root biomass decrease linearly with distance from the center of the shrub crown? Another example for lines 351-352: how was the photodegradation coefficient calculated from the mass-loss rate. I am also a bit confused as to why only certain parameters are included in Table 2 and not all (e.g. why is the photodegradation coefficient not included for example)?*

Response: That section (numbered as 2.4.2 in revised paper) has been checked and more information has been added. Photodegradation coefficient kp has been added to Table 2.

*Lines 377 and 379: I think the 2nd Q10() in equation 28 should be Q10($\vartheta$) and the same in equation 30? How did you come to define equation 31 in this way? Based on the aforementioned studies? Which method did you use to perform the fits shown in Figure 2 and equations 32 and 33?*

Response: The 2nd Q10 in equation 28 has been revised to Q10($\theta$). For equation 31, there are not available numerical descriptions on such an rain effect so far; therefore we decided to add one to the algorithm. This equation has been tested in sensitivity analysis (see test for parameter $n_p$), which shows this equation may not be an important source of uncertainty. Future modelling may also consider to simplify this algorithm (Line 615). Fittings in figure 2 (Fig. 3 in revised manuscript) were performed by Matlab curve-fitting toolbox. The information has been added to Line 416 and 423.

*Section 2.4.1 title should mention the meteorological forcing data. A shorter title could be "model set-up". Do you have a reference for the PECE method?*

Response: The title has been changed as suggested (Line 324). Like forward/backward Euler, PECE method can be found in many textbooks related to ordinary differential equations, e.g. Butcher, John C. (2003), Numerical Methods for Ordinary Differential Equations, New York: John Wiley & Sons, ISBN 978-0-471-96758-3. The reference has been added to the manuscript (Line 482).

*Lines502-504: How/why did you choose which parameters to include in your sensitivity analysis?*

Response: The reason to choose the tested parameters has been better discussed in section 2.5.3.

*Section 2.4.3: to avoid some confusion in the results later, I might have the parameter sensitivity as a separate test from the comparison between plant covered and interspace soil CO2 flux (so add a test 5). I think this would help to emphasize the importance of the impact of plant heterogeneity in the text, given its importance in the paper title. It might be useful for the reader to have a small table summarizing all the tests, which processes they include, which site they correspond to, what the observations are measuring etc.*

Response: This suggestion was taken with gratitude. We separated the contents as suggested, and organized the tests to two part: i) to demonstrate the roles of componential C fluxes in regulating surface efflux; and ii) to find out the differences between plant cover and interspace in terms of C flux rates and sensitivities. see Section 2.5.3-2.5.4.

*Results What is shown on the bottom of Fig4a? Is that precipitation? It might be worth smoothing your hourly curves with a moving average window so we can see the variability better I would put precipitation on the hourly time series plots in Figure 5 as well.*

Response: The figures 4 and 5 have been revised following the suggestions. 3-day moving average trends were used to show the temporal dynamics better.

*Lines 540-541: It would be good to give the RMSE of C1 above as well for a comparison.*

Response: The RMSE value was added to C1 in Line 551.

*Figure 6: I like the addition of the diurnal bias plots - they are very informative. I would put the same scale for all C1, C2 and C3 plots to enable an easier comparison between the tests.*

Response: The scales has been set to same in Fig. 5 and Fig.7.

*Lines 542-544: Looking at Figure 6b it seems to me that the pattern of diurnal biases has changed for C3 compared to C2 and C1? There is now a positive bias around noon and a negative bias in the mid-morning and afternoon? Why do you think this is?*

Response: Indeed, the pattern became different in C3 compared to C2 and C1. It is probably caused by biases from photosynthesis & photodegradation, which were introduced into system in Test 3. We have revised the description in the manuscript (Line 565-568).

*Table 3 caption: component fluxes. It would also be better to say "for areas with plant cover and without (interspace)"*

Response: Table 3 caption has been revised as suggested.

*Figure 7: You mention FS in the caption but FT in the text and figure legends. Also, you refer to net $CO_2$ sequestration by the biocrust in the legend – isn't this FB (or FCt), of have I misunderstood? It would be helpful to the reader to make sure all the abbreviations you use for the fluxes are uniformly used across the text and figures. In fact, I would suggest adding an extra table with all the component flux abbreviations and their long name/description, given that there are many. This may help the reader cross-reference between the figures, tables and text.*

Response: We agree there have been several place with unnecessary naming. We have revised all the symbols and names to keep them consistent. Please see in the revised figure (Fig. 8).

*Table 4: are all the values listed the % change in C flux after manipulation compared to the base flux, or are some of the +/- values a change in the absolute C flux magnitude? Please detail this in the table caption.*

Response: Yes, all values listed in Table 4 are % changes. This point has been clarified in figure footnote.

*Lines 578-583: I am a bit lost as to main message of the second part of the sensitivity analysis. What does the sensitivity analysis suggest about how important the parameters are? If changing the parameter values does not result in that much change to the fluxes, does that mean that parameter or even that process is not actually important for modelling the flux? How have you decided how much to change the parameter*

*values? Perhaps it would be good to explore their full range (between their upper and lower bounds) in a proper sensitivity analysis (e.g. using the Morris method) in order to determine the full impact of the parameter values.*

Response: Yes, parameter sensitivity was studied to understand which parameter is more important and more likely to be the main source of uncertainty, as we have many site-specific parameters. Those parameter have high sensitivity then need to be used with extra caution, when applying the system to another space-time. For site-specific parameters like Ts, θ, Mtot etc., we modified the values by the same degree (±10%), so that their effects on C fluxes are easy to compare. Moreover, if the changes in simulated C fluxes are much smaller than the changes in certain parameter, we consider the simulated C fluxes could be insensitive to that specific parameter. Therefore, this kind of tests is a bit different from the Morris method. On the other hand, "full" impact is difficult to define. Some parameters, e.g. root biomass, may vary by several folds from one collar to another (see Wang et al., 2015, Biogeosciences). Also an artificial parameter setting (e.g. assuming extreme values for many parameters) may be unrealistic in many cases (e.g. a combination of very low moisture content and very high root biomass). Therefore, performing an Morris analysis like suggested could be difficult, while the upper/lower bounds of parameters are unclear in a combination.

*Some of the results are repeated in the discussion. Given that the results section is very short, it might be better to merge the results and at least some of the discussion that is very pertinent to each particular result for each separate sub-section (e.g. validity of the results, ...) and separate out each with a sub-heading. That way the reader is not switching between different aspects of the modeling in the results before having to come back to consider the implications of these results in the discussion. It is encouraging that the authors are aware and detail all the caveats of their work; however, the manuscript might benefit from a shorter, more concise discussion, particularly given the methods section is also (necessarily) long. A brief summary of the missing features of the model such as is given in the conclusions may be enough with a few extra sentences and references. Be sure not to repeat sections of the introduction or results in the discussion, e.g. lines 726-744 is largely a repetition of context and results. I appreciate it is hard to keep the results and discussion separate, which is why I have suggested combining at least some parts of the discussion with the results in a "results and discussion" section. This would also help to reduce the length of the manuscript. Other more general parts of the discussion could be put in a final "Conclusions and future perspectives" section.*

Response: We greatly appreciate these advices. In order to better structure this part, we moved a part of the discussion on model validity into result section 3.1. Now discussion (i.e. section 4.1 and 4.2) has focused on answering the two questions we proposed. The model uncertainties and further needs of research are discussed in the final section 4.3. We think this information would be helpful for future studies on experiments and modeling.

*MINOR COMMENTS In general: CO2 production, not productions. Both some missing and unnecessary "the" in places (e.g. line 95, no 'The' is needed, and occasionally the plural of a word is used where it should not be (e.g. CO2 productions). Check the text carefully. Please could you explain this sentence more: "In dryland soils, the interactions between CO2 transport and water cycle could also be intensive, due to the commonly high salinity/alkalinity of soils."? What do you mean by intensive?*

Response: Appreciated. We revised the text as suggested, see Line 60-62.

*Line 104: "However, both models focus on the patterns at the regional-scale with very simplified ecosystem processes and neglect stand-scale heterogeneities of water energy budget, and have not yet been validated by field measurements." I would turn this sentence into a positive one to highlight what you will do to add to the field and incorporate that into your following paragraph. Something like switching this sentence to read "we will build on this work by including complex processes related to... . Furthermore, we have validated our new model with extensive field measurements..."*

Response: we have revised the text, see Line 107-120.

*Line 384: Sponseller, 2007 and Cable et al. references missing*

Response: Citation to Sponseller has been added. Cable et al. 2013 was removed.

*Lines 409-411: Do you mean NPP and not NP?*

Response: Yes, it should be NPP here (Line 420-422). Corrected.

*Line 429: litterfall Line 456: probability, not probably?*

Response: Corrected as suggested (Line 439 and 466).

*Line 464: "The model simulation employed half-hourly meteorological factors" "the model was run with half-hourly meteorological variables"*

Response: Corrected as suggested (Line 474).

*Line 496: "contributed to the soil CO2..."?*

Response: Corrected (Line 505).

*Line 506: "It was also studied" "Furthermore, we studied" Line 507: "regarding the" "due to"*

Response: This paragraph has been revised (section 2.5.3).

*Line 521: pronounced*

Response: Revised as suggested (Line 542).

*Line 703: "should" instead of "shall"*

Response: This paragraph has been revised.

*Lines 705-709 reads more like "Conclusions"*

Response: This paragraph has been re-organized.

Final Response to reviewer #2

*General comment:*
*This study concerns the development and application of a highly detailed physically based patch-scale land-atmosphere energy, water and carbon balance model for a semiarid ecosystem with high plant-interspace heterogeneity. The model represents an expansion of the model developed by Gong et al. (2016. Ag. Forest Met) that compared patch scale water and energy exchange into soil-plant C exchanges. The model represents most of the C stocks and fluxes that you expect to be relevant for dryland ecosystems, but which are not normally represented in ecosystem C models like photodegradation, biocrust photosynthesis and respiration, gas and liquid phase co2 transport, etc. The model is used to simulate bare and plant shaded surfaces as well as biocrust covered surfaces and compared to measurements. The model was shown to be very capable of accurately simulating measured soil temperatures, soil moisture and soil respiration(Rs). The main findings were that totalCO2 production in the soil and Rs could deviate substantially from one another due to root uptake, crust respiration and photosynthesis, and variations in CO2 dissolution, emphasizing the processes beyond heterotrophic and autotrophic respiration and highly heterogeneous nature of CO2 cycling in patchy ecosystems. These results shed some light on the importance of these other processes that are not commonly represented in ecosystem models as well as our ability to represent them in ecosystem models. The paper is well written. The authors do a great job in discussing the background literature in the Introduction as well as tying their findings to previous studies in the discussion. The paper is very long, but this should probably be expected given the highly detailed modeling work that is being presented. Overall, there is nothing fundamentally flawed with the paper and I expect that this work will be of interest to ecosystem modelers, particularly those interested in dryland ecosystems.*

*My main complaint about the paper involves equifinality of the model results and the lack of data to be able to validate their findings on the relative roles of the different component fluxes. A model with far fewer parameters and processes is likely to be equally as capable of simulating soil moisture, temperature, and Rs for these cases or tests so how can one have much confidence that extra capabilities of the model (to represent the individual fluxes and transports like photorespiration, crust photosynthesis/respiration, CO2 uptake by roots) are valid? Table 3 is great, but it could be entirely fictitious. While I'm excited to see models being built with these processes considered, I'm wonder how we can build confidence that they are any better than simple, more empirical models already out there.*

Response to general comment:

We sincerely appreciate the comments of reviewer. The manuscript has been revised in light of the comments, with several main changes in the structure, figures and tables. All the revised texts have been marked in red color.

For the complaint on equalfinality, we fully are aware that over-complexity and over-parameterization could be important sources of uncertainties for process-based models; and that also, a well-trained empirical model with less parameters may still yield good fitting, for environmental conditions assumed not to change. In this work, we have aimed to understand the role of componental fluxes under varying environmental conditions.

For the complaint on lack of data support, we are aware that only one chamber for each step may not be enough, and fluxes like photorespiration and $CO_2$ removed by root or lateral flows are still lack of support from field data. These uncertainties, along with several other possible aspects, have been discussed in revised section 4.3.

*"A couple of relatively minor issues: 1. There could be better setup in the introduction. What are the objectives and rationale of this study? Major questions or hypotheses? "*

Response to minor issue 1:

This issue is pointed out by both reviewers and we have re-organized the manuscript to answer two specific objectives: i) to investigate the roles of componential C processes in regulating soil $CO_2$ effluxes in the studied ecosystem, and ii) to analyze the plant-interspace differences in the componential C processes. These issues has been discussed in introduction (Line 111-120). The results (section 3.2, 3.3) and discussion (section 4.1 and 4.2) have been re-organized accordingly.

"2. Figure 8 used as an estimate of photorespiration. How do you separate the effects of greenhouse effect under the clear chamber versus the shade effect of the opaque one? In other words, the opaque chamber shields the surface and reduces the heating when the chamber is closed. The clear chamber, by allowing solar radiation in and blocking thermal radiation out, is going to be heated much more potentially during the measurement cycle, potentially increasing heterotrophic respiration. Is Rs higher because of higher temps or because of photorespiration? "

Response to minor issue 2:

The C fluxes are measured by automatic chambers, which only seals the collar during measurement (2.5 minutes), then move away from the collar. Therefore, the collar was not blocked by chamber in most of time, and the temperature disturbance by measurement are marginal. As suggested by Figure 8, flux signals during the daytime nearly doubled during those periods; it is difficult to be explained as heating effect, as even 2 degree heating lead to < 10% changes in efflux (Table 4, sensitivity analysis). Also, the period was dry and with almost no rain event (see Fig. 6 in revised manuscript). Therefore, photorespiration by crust organisms is also unlikely.

"Text specific comments: L13. This sentence is an unusual way to open up an Abstract. I am wondering if it could be replaced with a sentence that provides context and rationale for the study "

Response: The abstract opening has been revised (Line 13-17), based on the revised objectives and contents.

L54. cannot

Response: Revised to "may not" (Line 57).

L55. periods

Response: Revised to "periods" (Line 58) as suggested.

L58 intensive? Also, why would water and CO2 transport be more intensive in the drylands?

Response: We are agree that this sentence is unclear. It has been revised to "In dryland soils of high salinity/alkalinity, $CO_2$ transport and water cycle are tightly coupled, as large inorganic C fluxes can be driven solely by dissolution and infiltration of $CO_2$ and carbonates" (Line 60).

L62. Here's another paper with the relevance of abiotic C with fluxes on the diurnal time scale. Hamerlynck, Erik P., et al. "Nocturnal soil $CO_2$ uptake and its relationship to subsurface soil and ecosystem carbon fluxes in a Chihuahuan Desert shrubland." Journal of Geophysical Research: Biogeosciences 118.4 (2013): 1593-1603.. There are several papers out that seem to show that inorganic C uptake is unlikely to be a very significant flux...see e.g. review in Schlesinger, William H. "An evaluation of abiotic carbon sinks in deserts." Global change biology 23.1 (2017): 25-27.

Response: It is true that inorganic C uptake may not be very significant flux in many cases. Our simulation also showed that such a flux was only about 15% of total emission from collar, and those C may still emit somewhere during the transport and have even less impact at ecosystem level. The main idea to include the transport processes are to better explain the variations of point-scale efflux, which may not necessarily link to changes in total soil C pool, but are just noises caused by transportation processes.

L69. matter

Response: Revised as suggested (Line 71).

L70. "could maintain inactive"?

Response: Revised to "could be inactive" (Line 72).

L77. Might consider H. Throop's work here, e.g., Throop, Heather L., and Steven R. Archer. "Resolving the dryland decomposition conundrum: some new perspectives on potential drivers." Progress in botany. Springer Berlin Heidelberg, 2009. 171-194. If you can't find this chapter, she has several articles about photodegradation.

Response: Good suggestion. Citation of Throop et al. 2009 has been added (Line 80).

L79. periods

Response: Revised as suggested (Line 81)

L95 Define "global change"

Response: Revised to "global climate change" (Line 97)

L104. "works"

Response: Revised as suggested (Line 106)

L114-115. This sentence seems out of place. If this represents an advance of Gong et al. you should cover what this model development is.

Response: This paragraph has been revised to better state the aims and objectives of this study and the advance of this modeling (Line 111-120).

L118. How about "Model Overview"

Response: Revised as suggested (Line 124).

L119. modeling work was based on measurements?

Response: Reworded as "... model was build based on ... " (Line 125)

L128. Don't understand this sentence

Response: The sentence has been reworded (Line 134).

L456 "probably"?

Response: Revised as suggested (Line 466).

L491. Later on, Test 4 is mentioned, but it should probably be included in this paragraph

Response: Test 1-3 are for model validation, Test 4-5 are for sensitivity analysis, so we consider that it would be better to separate them into different sections.

L503. component

Response: This paragraph has been revised (section 2.5.3).

L521. "was more pronounced"

Response: Revised as suggested (Line 542).

L523-525. I don't know of many soil C3 water probes that are good at measuring frozen water content. Are you sure the measurements are valid during these times?

Response: It is true that water content measurement during freezing period may not be reliable. We have changed the statement here (Line 543-544).

L534 4b?

Response: revised to 5b (Line 554).

L554. All the variables need to be clearly redefined in the Table caption so that this paragraph is much easier to understand.

Response: Table 3 has been revised and the definition of variables has been added to footnote

L565. Compared to

Response: The paragraph has been revised (Line 587).

L566 ,irrespective of the size...

Response: Revised as suggested (Line 588).

L573. "compared"

Response: Revised as suggested (Line 604).

L598. "our model capably reproduced the time series for the water and energy fluxes at ..."

Response: Revised as suggested (Line 547).

L605 Suggest using another heading before this paragraph, something like "modeling uncertainties"

Response: This has been suggested by both reviewer and we have reorganized this part to section 4.3

L705. provides L708 caution L709 Our simulations showed that a

Response: The result section has been reorganized. Those sentences has been rewritten or removed.

L773. uptake

Response: revised as suggested (Line 792).

L785 Are the model and data available for others to use ?

Response: So far, it is distributed among several collaborators but yes. We are still trying to include the aboveground vegetation before wider distribution.

Table 3. All terms need to be defined in this table caption including Fs, Fft PCt

Response: Revised. Definitions have been added to table footnote.

Figure. 1. The photo is really too small to see much of anything. Suggest deleting this so there is more space for the conceptual figure

Response: Revised. The conceptual framework has been displayed separately as Fig. 2.

Fig. 4. Ppt is not labeled or given a scale

Response: Revised (see Fig. 5).

Fig. 6. Greek letter is not defined in the caption.

Response: Revised (see Fig. 7).

Fig. 7. This figure is very hard to see. Could you use more colors for the different symbols so that it is easier to see?

Response: We revised the figure by differentiating the coloring (Figure 8). The resolution of initial figure was also limited by the file-size restrictions of discussion paper. We will upload bigger images with better qualities in the final submission.

.

[revised manuscript text omitted]
 plant-covered and interspace areas, respectively. A positive percentage (inside bracket) indicates a greater sensitivity ($|dF|$) of the flux at plant cover than interspace, whereas a negative value indicates a lower sensitivity. Definitions of fluxes and sensitivities see Table 3 and Section 2.5.3. [c] the difference in sensitivity are smaller than 0.1% by value.

Figures

[Figure]

Figure 1. Site position (a), overlook of measured ecosystem (b), appearance of soil surface at collar C1 (c), C2 (d) and C3 (e), and layout of representative land unit (RLU, adopted from Gong et al., 2016)

[Figure]

Figure 2. Conceptual framework of process-based modelling. Solid arrows represent flows of masses and dash arrows represent flows of information.

[Figure]

Figure 3. Measured and fitted bulk respiration (a) and photosynthesis (b) of the lichen topcrust as functions of temperature and water content.

[Figure]

Figure 4. Measured and modelled soil temperature (a) and soil moisture content (b) at 10 cm depth for

$F_S$ site, and as compared to the EC site in year 2013 by Gong et al. (2016).

[Figure]

Figure 5. Measured and modelled hourly $F_S$ for non-crusted soil (a), the temporal pattern of the bias of simulated hourly $F_S$ (b) and the comparison of measured and modelled daily $F_S$ (c) during 2013-2014.

[Figure]

Figure 6. Measured and modelled $F_S$ of lichen-crusted soils for opaque (a, c) and transparent chambers (b, d) at hourly (a, b) and daily (c, d) scales during 2013-2014.

[Figure]

Figure 7. Diurnal patterns of biases ($\zeta$) in the simulated hourly $F_S$ for lichen-crusted soils using opaque (a) and transparent chambers (b), and the cumulative probability of the biases during wetting and drying periods (c) during 2013-2014. The wetting period included the raining days and a 1-day period after each rainfall. The drying period included the rest time of the years other than the wetting period.

[Figure]

Figure 8. Simulated component $CO_2$ exchanges by biocrust and root-zone soil (a), the simulated $CO_2$

fluxes before and after example rain events of 2.3 mm (b), 7.6 mm (c) and 12.8 mm (d) sizes, and the comparison of $F_T$ and $R_R$ during wetting and drying periods during 2013-2014. The wetting period included the raining days and a 1-day period after each rainfall. The drying period included the rest time of the years other than the wetting period.

[Figure]

Figure 9. Comparison of the measured $F_S$ from lichen-crusted surfaces using opaque and transparent chambers during a dry period (day 83-103) in spring 2013.

---

## Author Response (AR2)

*Comments to the Author:*

*The authors did a good job in revising the manuscript in line with the concerns of the reviewers. Nevertheless, the manuscript requires further minor revisions before becoming acceptable for publication:*

*(1) the comment about equifinality (reviewer 2) echoed by the handling editor in the initial decision was not addressed in the revised manuscript. This is still an outstanding concern. "The comment about equifinality (reviewer 2) is an important comment and should be discussed in a separate paragraph."*

Response to comment (1):

We have addressed the question, and used the concept of equifinality as a general term to lead the discussion about model uncertainties (Section 4.3, first paragraph). Two related citations have been added to the reference list (L838-841).

*(2) Ask a native English speaker (preferably a colleague in the lab) to check the grammar of the manuscript. The manuscript can be understood but there are several (small) problems with for example the use of the word "the" and "a", tenses of verbs, … Carefully check the manuscript for typos and small mistakes. I found many and listed few below. This kind of problems require the reader to read a sentence more than once before it could be correctly understood which reduces the chances that the reader will use and cite your work.*

Response to comment (2):

We have revised thoroughly the text for these grammar corrections. The corrected places are all marked in red color in the revised manuscript.

*Examples of edits that would improve the readability of the manuscript:*

*Start the abstract with a sentence that gives the context of the study. Try not to make the abstract any longer than it currently is.*

Response:

We have revised the beginning of abstract (L13).

*L110-120 should be moved to the method section (between L123 and L124) or should be deleted.*

Response:

This content has been removed as suggested.

*L157 Mention that the model is then used to answer scientific question. Give the question.*

Response:

The studied questions have been addressed here, see L143-145.

*L468 Leech should be replaced by leach*

Response:

Revised as suggested (L454).

*L549. Define qualitative labels such "reasonably" or even better don't use them at all and leave it to the reader to decide whether they find this is a poor, reasonable or good fit. L549 could be deleted. Check the rest of the manuscript for undefined qualitative labels, for example, L573 "well".*

Response:

These lines and expressions have been revised as suggested (L535, 558).

*L551 Replace umol by µmol.*

Response:

The units are revised (L392, 536, 537, 547).

*L559 Mention the statistical test that was used to test the significance of this increase. If there was no statistical test use "substantially" instead of "significantly".*

Response:

The description has been revised to "substantially" (L663).

*L610 Should the r in Mr be written as a subscript instead of superscript?*

Response:

The R letter should be superscript, following the definition in equation (14).